# National forest carbon harvesting and allocation dataset for the period 2003 to 2018

Daju Wang[1], Peiyang Ren[1], Xiaosheng Xia[1], Lei Fan[2], Zhangcai Qin[1], Xiuzhi Chen[1], Wenping Yuan[1]

[1]School of Atmospheric Sciences, Guangdong Province Data Center of Terrestrial and Marine Ecosystems Carbon Cycle, Sun Yat-sen University, Zhuhai, Guangdong 510245, China.
[2]Chongqing Jinfo Mountain Karst Ecosystem National Observation and Research Station, School of Geographical Sciences, Southwest University, Chongqing 400715, China

*Correspondence to*: Wenping Yuan (yuanwp3@mail.sysu.edu.cn)

**Abstract.** Forest harvesting is one of the anthropogenic activities that most significantly affect the carbon budget of forests. However, the absence of explicit spatial information on harvested carbon poses a huge challenge in assessing forest harvesting impacts, as well as the forest carbon budget. This study utilized provincial-level statistical data on wood harvest, the tree cover loss (TCL) dataset, and a satellite-based vegetation index to develop a Long-term harvEst and Allocation of Forest Biomass (LEAF) dataset. The aim was to provide the spatial location of forest harvesting with a spatial resolution of 30 m and quantify the post-harvest carbon dynamics. The validations against the surveyed forest harvesting at 133 cities and counties indicated a good performance of the LEAF dataset in capturing the spatial variation of harvested carbon, with a coefficient of determination ($R^2$) of 0.83 between the identified and surveyed harvested carbon. The linear regression slope was up to 0.99. Averaged from 2003 to 2018, forest harvesting removed $68.3\pm9.3$ Mt C $yr^{-1}$, of which more than 80% was from selective logging. Of the harvested carbon, $19.6\pm4.0\%$, $2.1\pm1.1\%$, $35.5\pm12.6\%$ $6.2\pm0.3\%$, $17.5\pm0.9\%$, and $19.1\pm9.8\%$ entered the fuelwood, paper and paperboard, wood-based panels, solid wooden furniture, structural constructions, and residues pools, respectively. Direct combustion of fuelwood was the primary source of carbon emissions after wood harvest. However, carbon can be stored in wood products for a long time, and by 2100, almost 40% of the harvested carbon during the study period will still be retained. This dataset is expected to provide a foundation and reference for estimating the forestry and national carbon budgets. The 30 m × 30 m harvested carbon dataset from forests in China can be downloaded at https://doi.org/10.6084/m9.figshare.23641164.v2 (Wang et al., 2023).

## 1. Introduction

As a critical terrestrial ecosystem type, forests play a pivotal role in regulating the global carbon cycle (Dixon et al., 1994). Over the period 2001 to 2019, global forests sequestrated 2.07 Gt C yr$^{-1}$ from the atmosphere (Harris et al., 2021), contributing more than 70% of the global terrestrial carbon sinks (Friedlingstein et al., 2022). Forests also provide numerous important ecosystem services to society (Costanza et al., 1997), which may substantially impact the forests carbon sink (Lal et al., 2013). One of

the most important services is the provision of wood products (Deal and White, 2012), which results in the transference of carbon from forest ecosystems into the social system. The carbon harvested from forests is typically transferred into the pools of Harvested Wood Products (HWPs) (the full names and abbreviations of terminologies are listed in Table S1), through their usage of wood for constructions, furniture, and fine papers, among others. These products are eventually decomposed and emitted into the

atmosphere with different turnover times (IPCC, 2014, 2019a). For example, when harvested wood is used to make paper, the carbon within the paper decomposes and returns to the atmosphere within several years (Brunet-Navarro et al., 2017). On the contrary, the harvested carbon entering the wood used for constructions has a slow turnover rate and can be stored for many years (Brunet-Navarro et al., 2017). Previous studies have highlighted the large uncertainties in estimating carbon emissions of wood post-

harvest due to a lack of information about the proportion of wood entering various wood pools (Skog et al., 2004).

      In China, forests are playing an increasingly important role in terrestrial carbon sinks benefiting from long-term afforestation projects (Liu et al., 2014). A recent study based on satellite data revealed that China is one of the few countries experiencing persistent greening over the past decades (Chen et al.,

2019; Yuan et al., 2019). Unlike India, where croplands dominate greening, nearly half of China's greening comes from forests (Chen et al., 2019). However, despite these positive trends, China ranks as the second-largest timber consumer and the largest wood importer globally (Research and Market, 2019). In 2017, the consumption of wood in China reached 192.5 million cubic meters, with 43.6% of that consumption being supplied by domestic harvests (Research and Market, 2019). A previous study has

highlighted that harvesting is the primary cause of forest disturbance in China, accounting for 85% of the average annual forest loss (Curtis et al., 2018). However, the impacts of forest harvesting on the terrestrial carbon cycle has not been estimated yet due to a lack of available data on harvested carbon. Accurate

forest harvesting data are crucial for measuring the national carbon sink-source balance, an important component of national carbon budget analysis related to forest resources and wood utilization (Winjum, et al., 1997).

Although some efforts have been made, there are still large uncertainties in estimating harvested carbon in forest ecosystems (Hurtt et al., 2011, 2020). For example, the Land-Use Harmonization 2 (LUH2) dataset provides annual harvested biomass carbon data from primary and secondary forests (Hurtt et al., 2011, 2020). However, the spatial resolution of the LUH2 dataset ($0.25° \times 0.25°$) is too coarse to analyze the dynamics of forest ecosystems at the regional or local scales, and its performance in assessing carbon harvest has not been examined in China. In addition, to our knowledge, no studies have been conducted to quantify the proportion of harvested wood or carbon entering various wood pools, which is crucial for estimating carbon emissions returning to the atmosphere (Skog et al., 2004; Johnston and Radeloff, 2019). Over the past decades, constructions, papermaking, and furniture manufacturing have shown substantial changes in China (Zhang et al., 2019; FAO, 2023), largely influencing the proportion of harvested carbon among various wood pools and the magnitude of emissions.

In this study, we developed a Long-term harvEst and Allocation of Forest Biomass (LEAF) dataset to provide: (1) the spatial location of harvested carbon with a high resolution of 30 m; (2) the proportion of harvested carbon allocated into various wood pools; and (3) the lagged carbon emissions of harvested carbon from the aforementioned wood pools. The tree cover loss (TCL) dataset developed by Hansen et al. (2013) and the interannual variation of a satellite-based vegetation index were used to determine the location of harvested carbon, and the provincial statistical biomass storage provided by the China Forestry and Grassland Statistical Yearbook was used to quantify the magnitude of harvested carbon. Post-harvest carbon dynamics were estimated based on a first-order decay (FOD) function according to the post-harvest wood use provided by the statistical data (IPCC, 2014, 2019a).

## 2. Methods and Materials

This study aimed to generate a Long-term harvEst and Allocation of Forest Biomass (LEAF) dataset, which is a component of the Terrestrial Ecosystem Disturbance (TED) dataset, named as TED-LEAF. The LEAF dataset includes the location and magnitude of forest harvesting and the estimates of post-harvest carbon dynamics. The identification of forest harvesting was based on the detection of changes

in multi-temporal vegetation indices. Combined with statistical harvest data, the forest harvesting and other disturbances causing such changes could be separated. Utilizing the classification of HWPs provided by statistical data, we estimated the delayed carbon emissions by 2100 from HWPs based on IPCC methodologies with China-specific activity data.

**2.1 Method of calculating harvested carbon**

We aimed to identify two types of forest harvesting: clear-cutting and selective logging (Fig. 1). In this study, clear-cutting is the harvesting of an entire stand at once on a scale of 30 m × 30 m, while selective logging is the harvesting of a portion of the stand within that area that is suitable and should be harvested.

For clear-cutting, the location was determined using the TCL dataset produced by Hansen et al. (2013). The TCL dataset indicates stand replacement disturbance or the complete removal of tree cover canopy at a scale of 30 m × 30 m. It has been widely used to identify deforestation globally or in multiple regions (Hansen et al., 2013; Curtis et al., 2018). To calculate the harvested carbon from clear-cutting, it was necessary to determine the above-ground biomass carbon (AGB, t C) storage for each pixel. Only

the 9th National Forest Inventory (NFI) provided both provincial forest AGB density ($CD$, t C ha$^{-1}$) and forest area ($S$, ha). Then, we calculated pixel-level (30 m × 30 m) AGB storage for 2014 to 2018 as:

$$AGB_j = CD \times S \times \frac{NDVI_j}{SNDVI} \qquad (1)$$

where $AGB_j$ indicates the AGB of the $j$th forest pixel in a given province and year; $NDVI_j$ denotes the Normalized Difference Vegetation Index (NDVI) of the $j$th forest pixel in that province for that year;

$SNDVI$ is the sum of NDVI of all forest pixels in that province for that year. According to the province-level biomass storage (V, m$^3$) provided by the 6th to 8th NFIs covering 1998 to 2013 at 5-year intervals, this study used the following method to calculate pixel-level (30 m × 30 m) AGB storage for 2003 to 2013:

$$AGB_j = V \times \frac{NDVI_j}{SNDVI} \times Coef \qquad (2)$$

where $Coef$ (t C m$^{-3}$) is the coefficient that converts biomass storage (V) to biomass carbon (AGB). Combining the provincial forest biomass storage ($V_9$, m$^3$) from the 9th NFI, we calculated the provincial $Coef$ (Table S2) as:

$$Coef = \frac{CD \times S}{V_9} \qquad (3)$$

Then, the harvested carbon was calculated for all pixels corresponding to clear-cutting derived from the TCL dataset. The total harvested carbon from clear-cutting ($HC_C$, t C) in a given province can be calculated by aggregating all pixels occurring tree cover loss.

For selective logging, we developed a satellite-based method to identify the location and magnitude of selective logging. The method was based on the principle of multitemporal satellite-based vegetation index analysis and detected the changes of NDVI between two adjacent years. This approach relied on two fundamental assumptions. First, we assumed that NDVI values decreased resulting from selective logging. Therefore, we calculated the NDVI difference ($NDVI_{diff}$) between the current year ($NDVI_t$) and subsequent year ($NDVI_{t+1}$) at all pixels (Eq. (4)) and determined the possible logging locations with decreased $NDVI_{diff}$, where $NDVI_{diff} < 0$ indicated potential selective logging.

$$NDVI_{diff} = NDVI_{t+1} - NDVI_t \tag{4}$$

Second, we assumed that the reductions in NDVI values resulting from selective logging would be more significant compared to the decreases caused by other factors such as droughts, heatwaves, ice storms, and insect outbreaks (Yuan et al., 2014), without considering fires due to their low frequency in China (Curtis et al., 2018). Changes in vegetation coverage caused by logging activities were expected to have a drastic and rapid impact on ecosystems compared to other environmental changes and disturbances. Therefore, the decreased magnitude of NDVI values due to selective logging was assumed to be the largest. Based on this assumption, we can identify selective logging areas by focusing on pixels exhibiting larger $NDVI_{diff}$. For a specific province, all pixels with negative $NDVI_{diff}$ values were sorted by ascending order in $NDVI_{diff}$. The pixels at the front of the sorted list had more negative $NDVI_{diff}$ values and a larger likelihood of being logging locations. Thereby, a threshold of $NDVI_{diff}$ ($NDVI_{Th}$) was needed to distinguish selective logging from other disturbed pixels. Selective logging was considered to have occurred only when $NDVI_{diff} < NDVI_{Th}$. Then, the harvested carbon ($HC_S$, tC) was calculated in all pixels corresponding to selective logging according to Eq. (5):

$$HC_{s\,j} = \frac{NDVI_{diff\,j}}{NDVI_j} \times AGB_j \tag{5}$$

Therefore, the province-level statistical harvested carbon (SHC) was used to determine the $NDVI_{Th}$ of each province, which made the sum of HCc and HCs equal to SHC. Theoretically, the determined $NDVI_{Th}$ should be less than 0. However, in several provinces, when $NDVI_{Th}$ was set to 0, the sum of $HC_C$ and $HC_S$ (identified harvested carbon) was still lower than SHC (e.g., Anhui), implying that the identified

harvested carbon was underestimated. Then we assigned the unidentified harvested carbon (SHC−HC$_C$−HC$_S$) to pixels where selective logging occurred.

$$HC_{S1\,j} = (SHC - HC_C - HC_S) \times \frac{NDVI_{diff\,j}}{SNDVI_{diff}} \tag{6}$$

where $HC_{S1\,j}$ is the harvested carbon added by the $j$th pixel where selective logging occurred; $NDVI_{diff\,j}$ indicates the $NDVI_{diff}$ of the $j$th pixel, and $SNDVI_{diff}$ is the sum of $NDVI_{diff}$ of all pixels where selective logging occurred in that province. This approach increased the harvested carbon for all pixels where selective logging occurred, so we also examined whether the harvested carbon exceed their AGB at the pixel scale. If so, we counted these pixels as clear-cutting. To assess the accuracy and validity of the dataset, we additionally quantified the proportion of pixels with harvested carbon exceeding their AGB compared to the total number of harvested pixels at the province level.

In addition, this study assumed that forest harvesting did not occur in the National Nature Reserves and high-altitude regions because of high transport costs and harvesting expenses. For the Tibetan Plateau and Yunnan Province, we assumed no harvesting at elevations higher than 2,500 m. In the other provinces, harvesting was assumed to be absent at elevations higher than 1,500 m, according to Nabuurs et al., (2019).

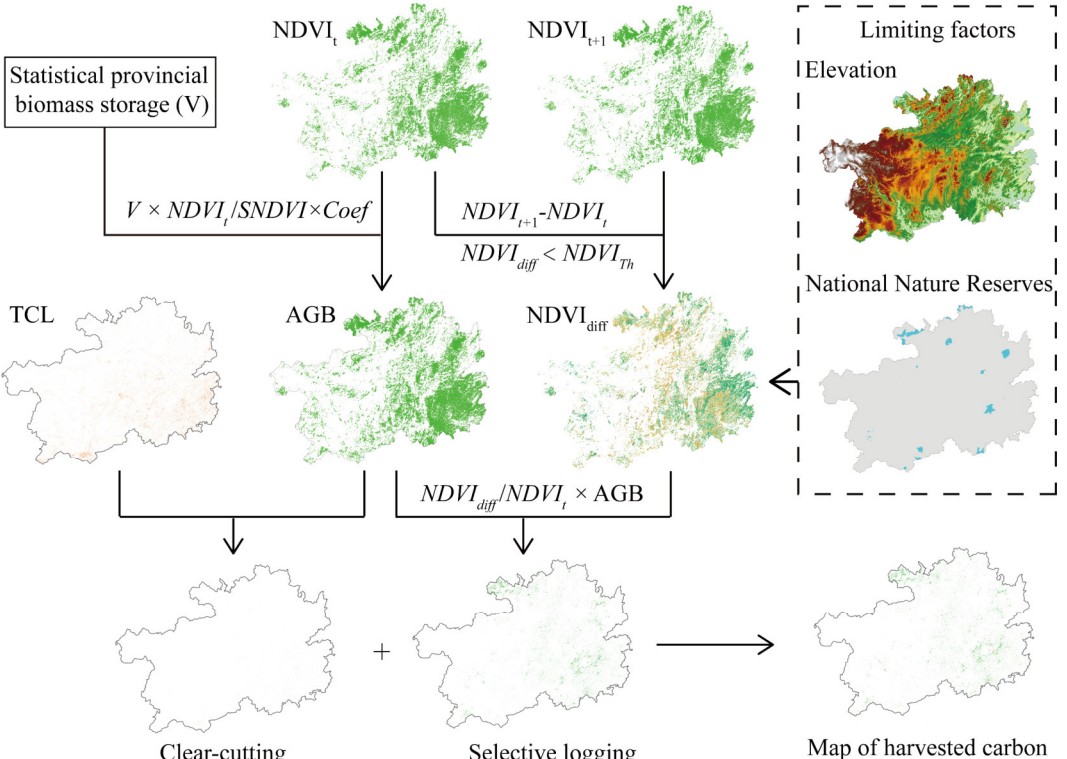

**Figure 1:** Flowchart for mapping forest carbon harvesting.

## 2.2 Annual carbon changes from HWPs in-use and end-use

The harvested wood was allocated into six wood pools (Fig. 2). (1) Fuelwood pool, the sum of commercial fuelwood and non-commercial fuelwood (i.e., farmers' fuelwood), where the wood is burned as fuel, resulting in immediate carbon emissions through combustion; (2) paper and paperboard pool, including household paper, printing paper, packaging paper, etc.; (3) wood-based panels pool, including plywood, fiberboard, particle board, and other wood-based panels, made from roundwood, wood residues (such as barks, branches, sawdust) or small stems bonded with adhesives, are commonly used as decorative panels for various applications like wall cladding and ceiling finishes; (4) solid wooden furniture pool, referring to solid wooden household items such as tables, chairs, wood beds, etc.; (5) structural constructions pool, referring to the structural components used to support buildings, such as beams, columns, and trusses; and (6) residues pool, including leaves, killed understory vegetation, and unutilized wood residues, which are typically left on the logging site or treated as fuel (Lippke et al., 2011; Stockmann et al., 2012), and were assumed as fuel in this study. The wood pools of (2), (3), (4), and (5) belong to HWPs pool, where the carbon will remain stored until the products are either retired from use or reach the end of their service life (Table S3) and are consequently discarded. Subsequently, these discarded products are then sent directly to landfill, where they decompose (Stockmann et al., 2012; IPCC, 2019a).

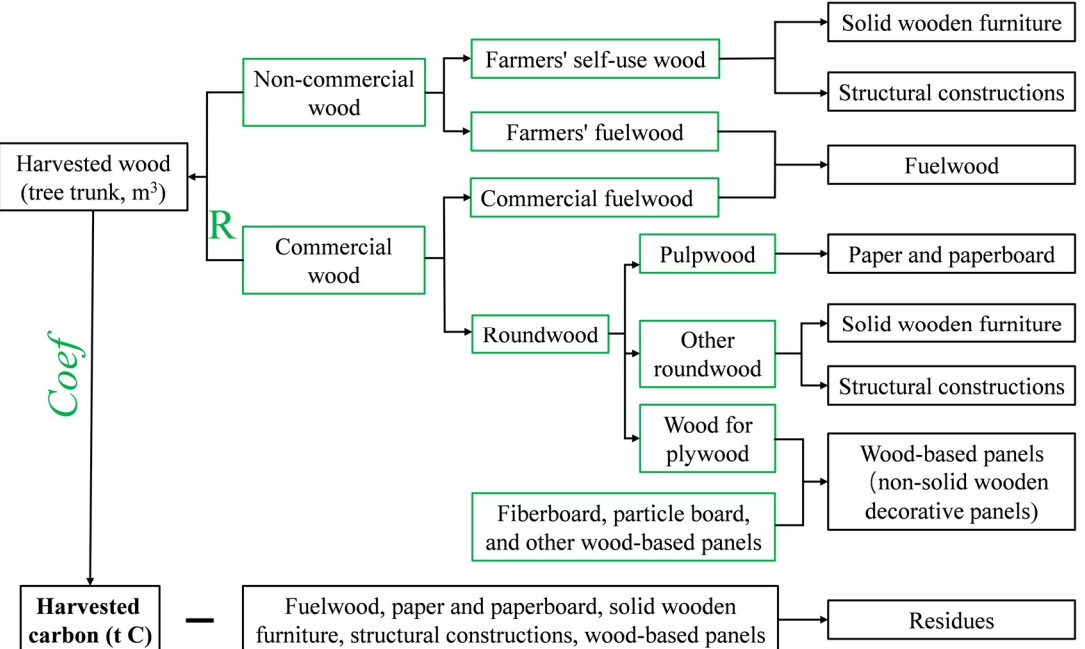

**Figure 2:** The allocation of post-harvest wood to the six wood pools. The green boxes indicate the variables are available from the China Forestry and Grassland Statistics Yearbook. R is wood output rate

of commercial wood, and $Coef$ (t C m$^{-3}$) the coefficient that converts harvested wood (m$^3$) to harvested carbon (t C) (i.e., biomass carbon of trunks, branches, leaves and understory vegetation).

In this study, the volume (m$^3$) of fuelwood, pulpwood (wood for paper and paperboard), wood-based panels, and the sum volume (m$^3$) of wood for solid wooden furniture and structural constructions can be obtained directly from the China Forestry Statistical Yearbook. The structural constructions and solid

wooden furniture pools were allocated as the percentage of 74.6±6.2% and 25.4±6.2% from their sum, respectively, according to China Timber and Wood Products Distribution Industry Yearbook. The wood in pools (1) to (5) was converted into carbon, with conversion factor of 0.229 for fuelwood and roundwood, and 0.269 for wood-based-panels. Then, the carbon entering the residues pool ($Residue_C$) can be calculated by subtracting the carbon in pools (1) to (5) from the total harvested carbon (SHC in

Eq. (21) of Sec. 2.3.2) as Eq. (7):

$$Residue_C = SHC - \left(W_{output} + F_{log}\right) \times 0.229 - wood\_based\ panels \times 0.269 \tag{7}$$

where $SHC$ (Eq. (21)) is the total harvested carbon for a given province, $W_{output}$ represents the volume of commercial wood output, and $F_{log}$ represents the volume of non-commercial wood. Unlike commercial wood, the actual logging volume of non-commercial wood can be used effectively by

195 farmers (Sec. 2.3.2). The sum of $W_{output}$ and $F_{log}$ is the total volume of pools (1) to (4), and wood-based panels here do not include plywood (Fig. 2).

### 2.2.1 Annual carbon changes in "HWPs in use"

Carbon stocks and decays from in-use HWPs were calculated using the methodologies for estimating carbon removal from HWPs described in the "2006 IPCC Guidelines for National Greenhouse

Gas Inventories" (IPCC, 2006a) and the "2013 Revised Supplementary Methods and Good Practice Guidance Arising from the Kyoto Protocol" ( IPCC, 2014). "Decay" in this paper refers to the discarding of HWPs from end uses (and sent to landfill), not biological decay. Moreover, here we focused on the carbon fate of wood harvested in China, the imports and exports were not considered. The FOD method (Pingoud and Wagner, 2006) was used to estimate the carbon change in the HWPs pool as:

$$C(i + 1) = e^{-k} \times C(i) + \left[\frac{(1 - e^{-k})}{k}\right] \times Inflow(i) \tag{8}$$

$$\Delta C(i) = C(i + 1) - C(i) \tag{9}$$

where $i$ denotes the year; $C(i)$ is the carbon stock in the HWPs pool at the beginning of year $i$; and $k$ ($k = ln(2)/HL$) is the decay constant (yr$^{-1}$). HL is the number of years it takes to lose half of the existing

material in the pool, which is a function of the country-specific service life of particular HWPs (HL=*service life* × *ln* (*2*)) (IPCC, 2019b). *k* for paper and paperboard, wood-based panels, solid wooden furniture, and structural constructions were calculated as 0.333, 0.125, 0.067, and 0.025, respectively. *Inflow*(*i*) is the inflow to the HWPs pool during year *i*, and Δ*C*(*i*) represents the carbon stock change of the HWPs pool during year *i*. Next, the amount of discarded organic carbon (*DOC*) deposited in the landfill, as the HWPs go out of use in year *i*, was calculated using Eq. (10):

$$DOC_i = Inflow(i) - \Delta C_i \tag{10}$$

## 2.2.2 Annual carbon changes in landfill

The *DOC* deposited in landfill was divided into three parts, (1) *DOC* for aerobic decomposition ($DOC_{ar}$), producing $CO_2$ until all available oxygen has been used up; (2) *DOC* for anaerobic decomposition ($DOC_{an}$), producing both $CH_4$ and $CO_2$; (3) *DOC* that will not be decomposed but stored long-term in the landfill ($DOC_{ls}$).

The $DOC_{ar}$ was calculated as:

$$DOC_{ar_i} = DOC_i \times f_{ar} \tag{11}$$

where $f_{ar}$ denotes the proportion of *DOC* undergoing aerobic decomposition. Based on the landfill situation in China, the $f_{ar}$ was assumed to be $0.28 \pm 0.15$ (Table S4). The $DOC_{ar_i}$ undergoes aerobic decomposition and releases as $CO_2$ in year *i*:

$$CO_{2\,ar_i} = DOC_{ar_i} \times \frac{44}{12} \tag{12}$$

The $DOC_{an}$ is calculated with Eq. (13), as:

$$DOC_{an_i} = DOC_i \times (1 - f_{ar}) \times DOC_f \tag{13}$$

where $DOC_f$ indicates the fraction of *DOC* that can be decomposed under anaerobic conditions (Table S5). The annual carbon change of $DOC_{an}$ in the landfill was calculated according to the FOD method (Pingoud and Wagner, 2006).

$$DOC_{an\,a_i} = DOC_{an_i} + DOC_{an\,a_{i-1}} \times e^{-k} \tag{14}$$

$$DOC_{an\,decomp_i} = DOC_{an\,a_{i-1}} \times (1 - e^{-k}) \tag{15}$$

where $DOC_{an\,a_i}$ represents the $DOC_{an}$ accumulated in the landfill at the end of year *i*; $DOC_{an\,decomp_i}$ denotes $DOC_{an}$ decomposed in the landfill in year *i*; the anaerobic decomposition generally occurs in the following year of deposition. *k* is the decay constant (yr$^{-1}$), estimated based on the environmental

conditions of landfill in China (Table S5). Then, the potentials to generate $CH_4$ and $CO_2$ from the anaerobic decomposition of $DOC_{an}$ were:

$$CH_{4_i} = DOC_{an\ decomp_i} \times F \times \frac{16}{12} \times (1 - R_T) \times (1 - OX_T) \tag{16}$$

$$CO_{2\ an_i} = \left[DOC_{an\ decomp_i} \times (1 - F)\right] \times \frac{44}{12} \tag{17}$$

where $CH_{4_i}$ and $CO_{2\ an_i}$ are $CH_4$ and $CO_2$ produced by anaerobic decomposition in landfill in year $i$. $F$ denotes the volume fraction of $CH_4$ in the generated landfill gas, with a value of $0.5\pm0.1$ (Cai et al., 2018). $R_T$ and $OX_T$ are the recovery rate and oxidation rate of $CH_4$ in China (Table S4). The total $CO_2$ decomposed from $DOC$ in year $i$ was the sum of $CO_{2\ ar_i}$ and $CO_{2\ an_i}$. The $CH_4$ emissions were transformed to $CO_2$ equivalents ($CO_2$e) to harmonize calculations of the overall global warming potential (IPCC, 2023).

The $DOC_{ls}$ can be calculated as:

$$DOC_{ls_i} = DOC_i - DOC_{ar_i} - DOC_{an_i} \tag{18}$$

Then, the carbon stock in the landfill at the end of year $i$ was calculated as:

$$DOC_{s_i} = \sum_0^i DOC_{ls_i} + DOC_{an\ a_i} \tag{19}$$

## 2.3 Datasets

### 2.3.1 National Forest Inventories

The NFIs provided nine periods of provincial biomass storage for 1973-1976, 1977-1981, 1984-1988, 1989-1993, 1994-1998, 1999-2003, 2004-2008, 2009-2013, and 2014-2018 (National Forestry and Grassland Administration, 2019). We used the biomass storage data during the 6th-9th NFIs to generate the distribution of AGB for 2003-2018.

### 2.3.2 Statistical forest harvesting data

The annual provincial wood output ($m^3$), extracted from the China Forestry and Grassland Statistical Yearbook, was categorized into two main types: commercial and non-commercial wood. Commercial wood includes fuelwood and roundwood, and roundwood was further divided into pulpwood, wood for plywood, and other roundwood (roundwood for directly use, internally processed roundwood, etc.). Non-commercial wood included farmers' self-use wood (i.e., the volume of wood logged by farmers for burning) and farmers' self-use wood (i.e., the volume of wood logged by farmers for their personal

consumption). Non-commercial wood refers to the actual logged volume, which can be totally used by

farmers. Commercial wood refers to wood output volume of peeled wood that meets the national wood

standards, not the actual logging volume. After wood logged, preliminary processing (such as peeling,

sawing, etc.) is carried out and some unusable or poor-quality wood is eliminated. Therefore, the

commercial wood output is generally less than the actual logged wood(National Forestry Administration,

2000). Based on the provincial wood output rate (R, i.e., the ratio of commercial wood output to wood

logged) provided by China's timber production plan from the National Bureau of Statistics (Table S2),

we calculated the actual annual wood logged for each province as:

$$W_{log} = \frac{W_{output}}{R} \tag{20}$$

where $W_{log}$ is the actual annual wood logged for producing commercial wood, $W_{output}$ is the volume of

commercial wood output. Then, the total harvested carbon (i.e., the SHC in Sect. 2.1) for a given province

was calculated as:

$$SHC = \left(W_{log} + F_{log}\right) \times Coef \tag{21}$$

where $F_{log}$ is the volume of non-commercial wood, and $Coef$ (t C m$^{-3}$) is the coefficient that converts

harvested wood (m$^3$) to harvested carbon (t C) (i.e., biomass carbon of trunks, branches, leaves and

understory vegetation) (Table S2).

**2.3.3 Surveyed forest harvesting data**

The surveyed forest harvesting data, i.e., the harvested volume at the city and county-level, was

retrieved from the official website of each provincial Forestry Bureau. To ensure the effective

implementation of the forest harvesting quota system, the provincial Forestry Bureau randomly selects

several cities and/or counties, conducting comprehensive field inspections of the annual forest harvesting.

The actual annual harvest volume of a specific city or county is determined through comprehensive cross-

validation of multiple methods, including declaration and approval records reviewing, on-site

measurements, and remote sensing monitoring, etc. These techniques are employed to ensure the

accuracy and reliability of the obtained harvest data. There were 133 records nationwide available from

2006 to 2018, concentrated in the provinces of Guizhou, Yunnan, Zhejiang, and Sichuan (Fig. S1).

**2.3.4 Forest cover map**

The Chinese Forest Cover Dataset, reconstructed by fusing the NFIs and twenty Land Use and Land

Cover datasets, was used as the forest cover base map. This forest cover accurately depicts the historical changes in China's forest cover in the period of 1980-2015, with overall accuracy from 76.9% to 99.4% (Xia et al., 2023).

### 2.3.5 Satellite-based vegetation index

The NDVI dataset from 2000 to 2020 calculated by Dong et al (2021) was applied to allocate AGB and identify selective logging. This NDVI dataset was calculated using all Landsat5/7/8 remote sensing data for the whole year, the NDVI maxima in each image year were obtained by data pre-processing and data smoothing (Dong et al., 2021).

### 2.3.6 Mask data

This study used the elevation and National Nature Reserves to exclude the regions where rarely occur harvesting. The elevation dataset was obtained from the Resource and Environment Science and Data Center of the Chinese Academy of Sciences (https://www.resdc.cn/) and the distribution of National Nature Reserves was provided by the National Earth System Science Data Center, National Science & Technology Infrastructure of China (http://www.geodata.cn).

### 2.4 Accuracy assessment

The surveyed forest harvesting data at city and county-level were employed to validate the accuracy of the LEAF dataset by conducting a comparison with the estimated harvested carbon. The relationships between the estimated harvested carbon and the corresponding surveyed harvested carbon were assessed by linear regression. Then the coefficient of determination ($R^2$) and slope of linear regression can be calculated, and the closer these two values are to 1, the better the estimates.

### 3. Results

### 3.1 Accuracy evaluation of LEAF dataset

Accuracy evaluation of the LEAF dataset generated from this study showed good performance in indicating spatial variations of harvested carbon in China. This study used province-level statistical harvested carbon to determine the threshold for identifying selective logging and used city and county-level surveyed harvested carbon ranging from 2006 to 2018 to examine the performance. The estimated

harvested carbon showed high consistency with surveyed harvested carbon, with $R^2$ of 0.83 and a linear

regression slope of 0.99 (Fig. 3).

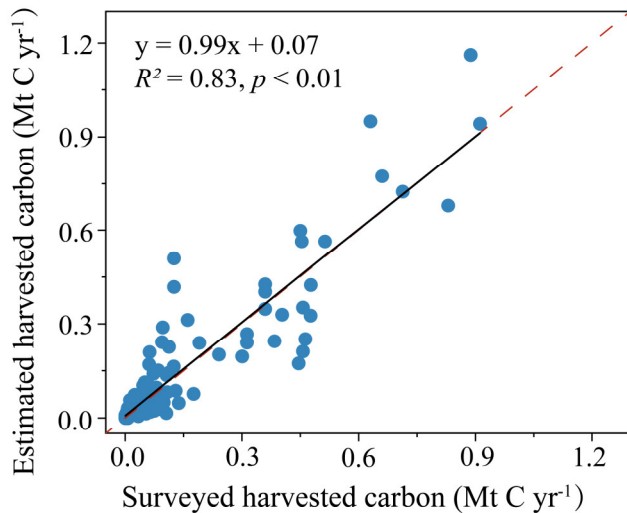

**Figure 3:** Comparison of estimated and surveyed harvested carbon from the official website of each provincial forestry bureau at the city and county levels across all investigated provinces. The solid line indicates the regression lines, and the red dashed line indicates the 1:1 line.

The validation also showed the performance of the LEAF dataset across various provinces and

periods. The dataset exhibited excellent skill in simulating the spatial variation of harvested carbon across

all surveyed provinces, with $R^2$ between estimates and surveyed data ranging from 0.24 to 0.96 (Fig. S2).

Zhejiang performed the best (Fig. S2e), while Sichuan's performance was relatively low, with $R^2$ less

than 0.5 (Fig. S2c). The LEAF dataset also effectively reproduced the spatial variations of harvested

carbon across different periods, with $R^2$ values ranging from 0.31 to 0.99 (Fig. S3). The estimates for

2013 and 2017 demonstrated superior performance, with both linear regression slopes approximating 1

(Fig. S3c and g). However, except for 2016, estimates from other periods displayed varying degrees of

underestimation or overestimation (Fig. S3).

**3.2 Spatial and temporal patterns of harvested carbon in China**

There was a large heterogeneity in harvested carbon over the regional scale. Harvesting mainly

occurred in Eastern and Southern China, and rarely in Northwest China (Fig. 4a and b). Guangxi Province

recorded the highest harvested carbon, constituting approximately 30% of China's annual total harvested

carbon. This volume is 2.5 times larger than that of Fujian, the province with the second highest harvested

carbon (Fig. 4c). In terms of harvesting ways, over 80% of the nationwide clear-cutting harvested carbon

originated from Guangxi, Guangdong, Fujian, Yunnan, and Jiangxi (Fig. 4c). Forest harvesting in Jiangxi,

Guangdong, Yunnan, and Hainan was dominated by clear-cutting, in these provinces, harvested carbon from clear-cutting comprised more than 50% of the total harvested carbon (Fig. 4c, Fig. 5a). For other provinces, selective logging was the main way of forest harvesting (Fig. 5b).

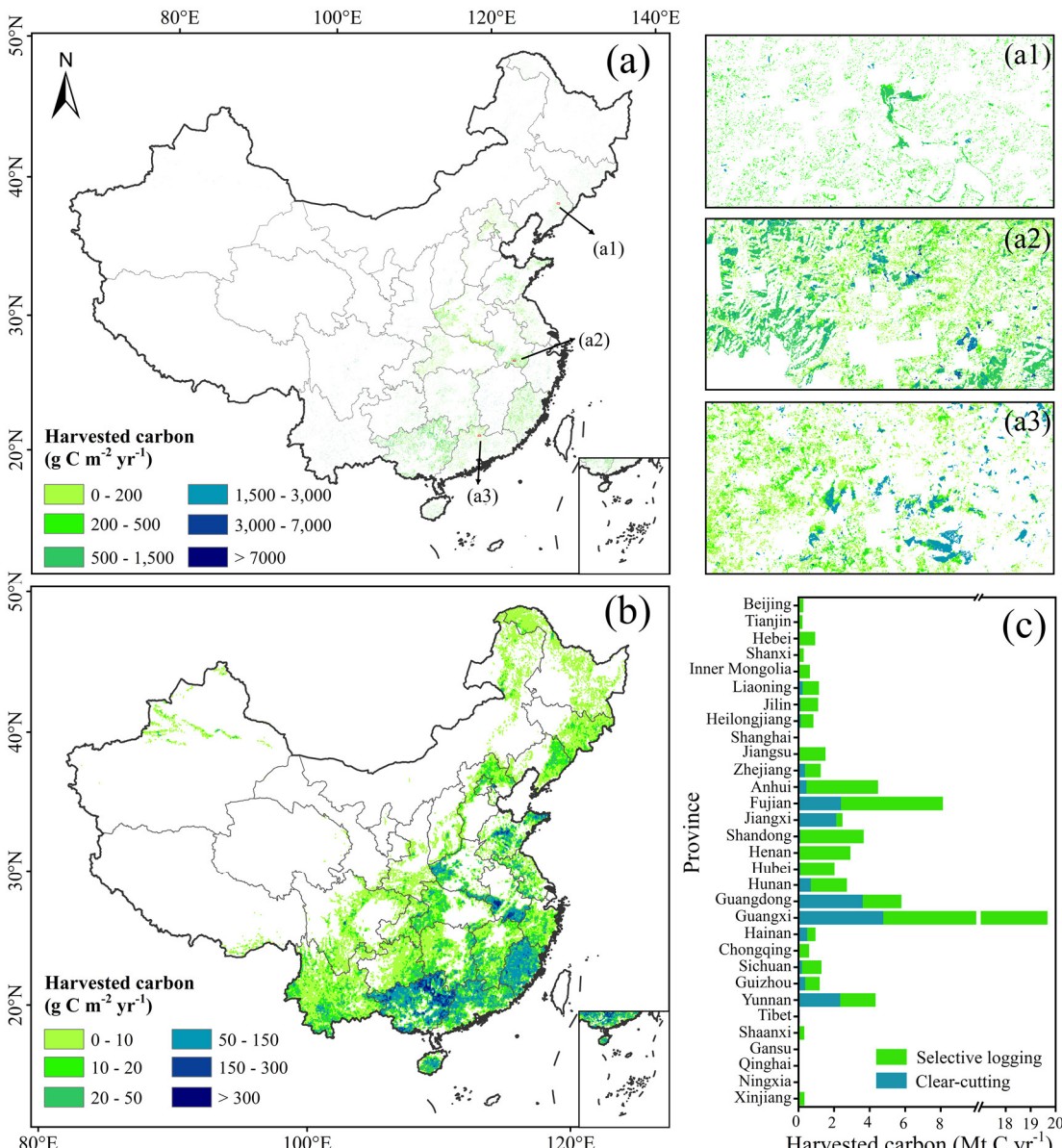

**Figure 4:** Map of forest harvested carbon for China in 2016 at (a) 30 m and (b) 0.1°resolution and the zoomed-in view of the example areas of (a) (a1, a2, and a3), the map at 0.1° was derived from a 30 m data upscaling, and (c) shows the harvested carbon from clear-cutting and selective logging by province in 2016.

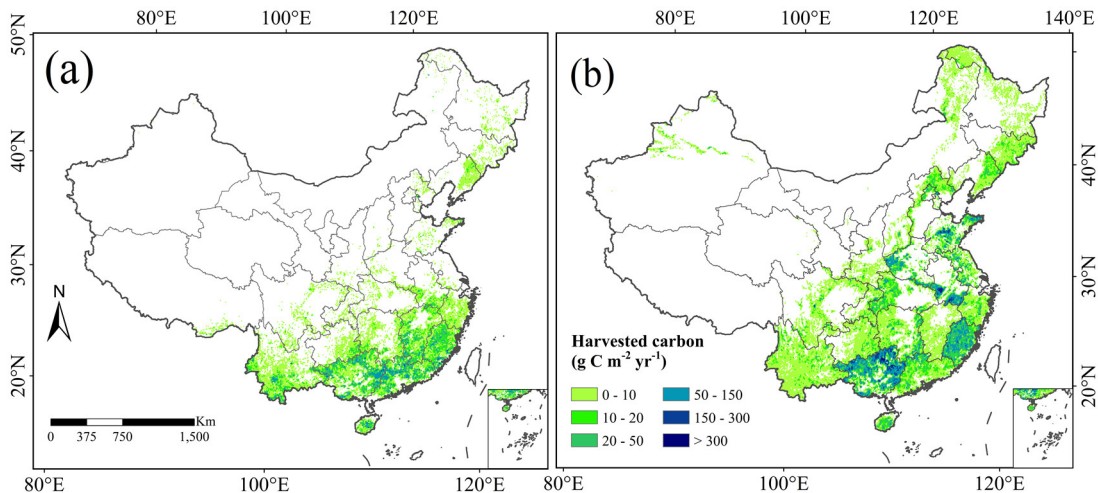

**Figure 5:** Map of forest harvested carbon from (a) clear-cutting and (b) selective logging for China in 2016 at 0.1°resolution.

China experienced an overall upward trend in forest carbon harvesting, with average harvesting of $68.3\pm9.3$ Mt C yr$^{-1}$ from 2003 to 2018 (Fig. 6). The most substantial growth in harvested carbon was observed from 2003 to 2008, increasing approximately 1.6 times from 49.8 Mt C yr$^{-1}$ in 2003 to 79.5 Mt C yr$^{-1}$ in 2008 (Fig. 6). Among all surveyed years, the harvested carbon peaked in 2008, primarily driven by an uptick in fuelwood harvest (Fig. 7). In subsequent years, the harvested carbon generally plateaued, with minor fluctuations. Selective logging was the main way of forest harvesting (Fig. 6). Averaged from 2003 to 2018, there was $55.6\pm7.2$ Mt C yr$^{-1}$ from selective logging, accounting for around 82% of the total harvested carbon (Fig. 6). Nevertheless, the proportion of harvested carbon from clear-cutting was overall on the rise, reaching its peak in 2012 when it constituted 30% of the total harvested carbon for that year (Fig. 6).

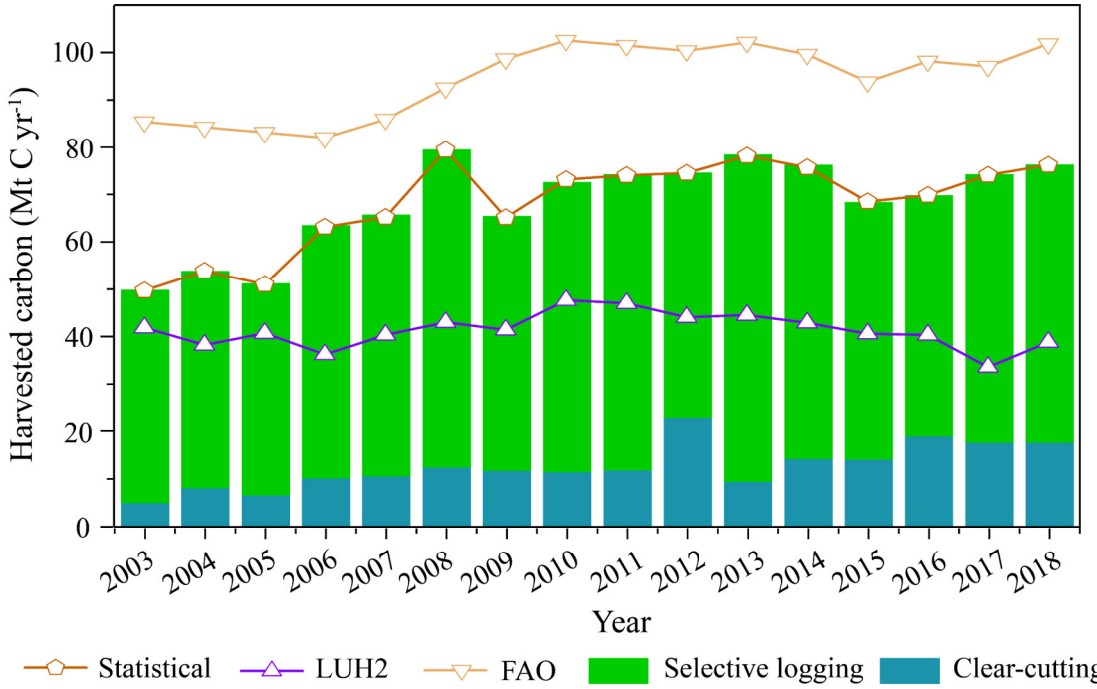

**Figure 6:** Long-term changes of harvested carbon in China between 2003 and 2018 from statistical, LUH2, FAO, and LEAF datasets.

### 3.3 Allocation of harvested carbon into the various wood pools

This study also provides the allocation of harvested carbon into the various wood pools with different lifetimes, including fuelwood pool, paper and paperboard pool, wood-based panels pool, solid wooden furniture pool, structural constructions pool, and residues pool (Sect. 2.2). Averaged from 2003 to 2018, over the entire China, harvested carbon was allocated into six pools with $19.6\pm4.0\%$, $2.1\pm1.1\%$, $35.5\pm12.6\%$, $6.2\pm0.3\%$, $17.5\pm0.9\%$, and $19.1\pm9.8\%$, respectively (Fig. 7). Wood for HWPs

mainly came from Southern China, with Guangxi contributing more than 20% of the national wood for HWPs annually, followed by Fujian (7.5%). Meanwhile, Guangxi also contributed up to 45.5% of the wood used for paper production (Fig. 8). Over 50% of the wood in Fujian was used for fuelwood, contributing nearly 30% of the national fuelwood. Similar to Fujian, more than 50% of the wood in Yunnan was used for fuelwood, making Yunnan the second-largest province in terms of wood supply for

fuelwood (Fig. 8). From 2003 to 2018, wood harvesting has been increasing, while wood for fuel has been decreasing, with its share decreasing from 29% to 13%. More wood was used for HWPs production. The wood residues decreased from 2003 to 2018 (Fig. 7).

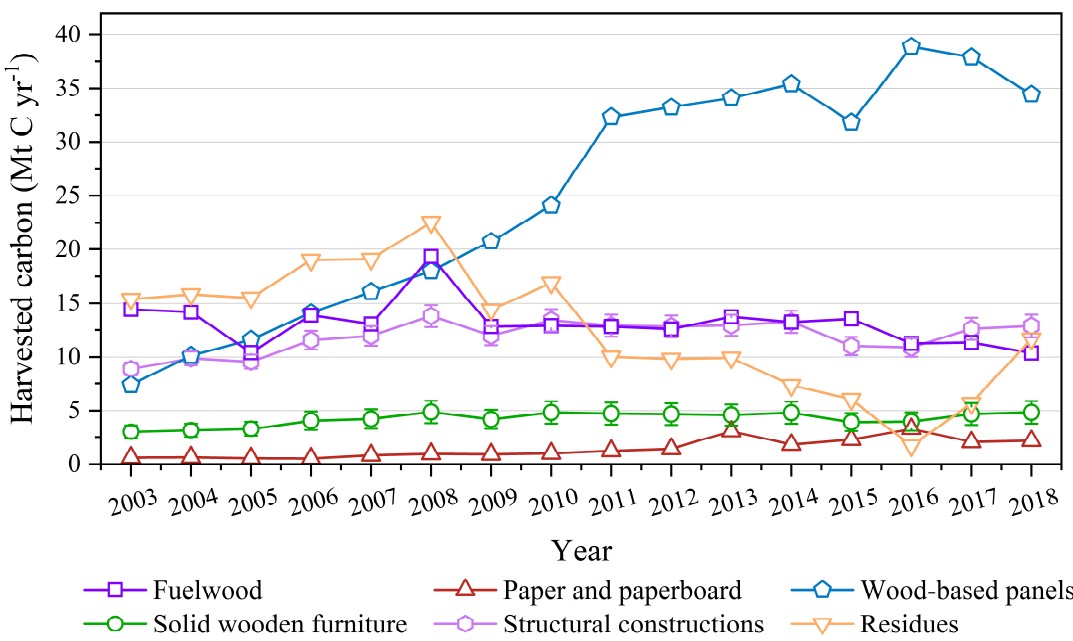

**Figure 7:** Long-term changes of different pools of harvested carbon from 2003 to 2018 in China.

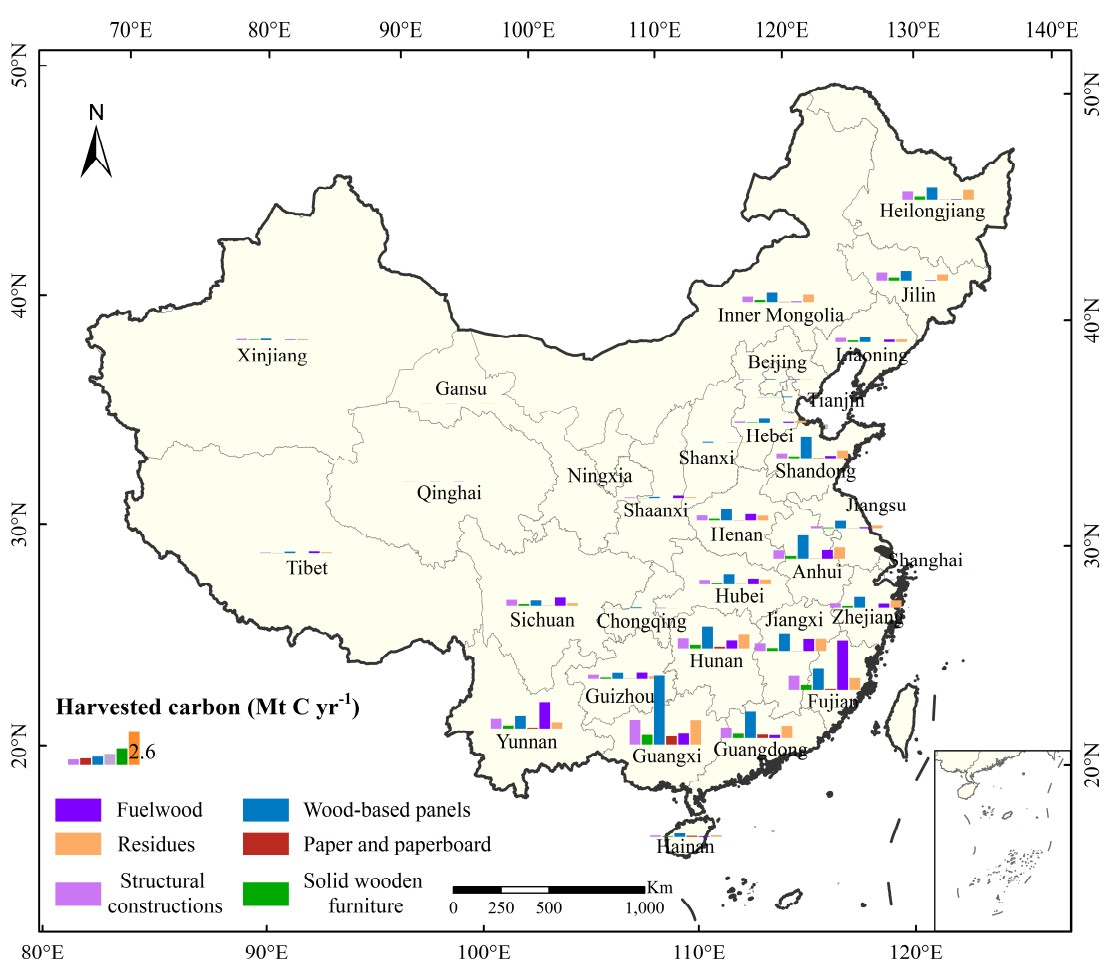

**Figure 8:** Pools of provincial harvested carbon averaged from 2003 to 2018.

We further quantified the delayed emissions of harvested carbon in various pools. Combustion of

fuelwood resulted in major carbon emissions, and wood-based panels contributed a major share of carbon stock (Fig. 9). Fuelwood was typically burned for a short period of time (1 year), resulting in average emissions of 48.1±7.4 Mt $CO_2e$ $yr^{-1}$ from 2003-2018 (Fig. 9c). Additionally, discarded HWPs that enter landfill undergo gradual decomposition, releasing an average of 3.1±3.0 Mt $CO_2e$ $yr^{-1}$, 14.4±12.2 Mt $CO_2e$ $yr^{-1}$, 2.5±1.8 Mt $CO_2e$ $yr^{-1}$, and 6.5±3.3 Mt $CO_2e$ $yr^{-1}$ for paper and paperboard, wood-based panels, solid wooden furniture, and structural constructions, respectively, from 2003 to 2100 (Fig. 9d). The total carbon stock sustained an upward trend, peaking at 2126.9±244.9 Mt $CO_2e$ in 2018 (Fig. 9a, b). As products were discarded and/or retired, the carbon stock in currently used products persistently declines without ongoing wood inputs (Fig. 9a). Furthermore, most of the waste products that enter landfill do not decompose and become a permanent carbon sink (Fig. 9b). About 39.2%±10.0% of harvested carbon during 2003-2018 will still be stored in the HWPs by 2100 (Fig. 9b).

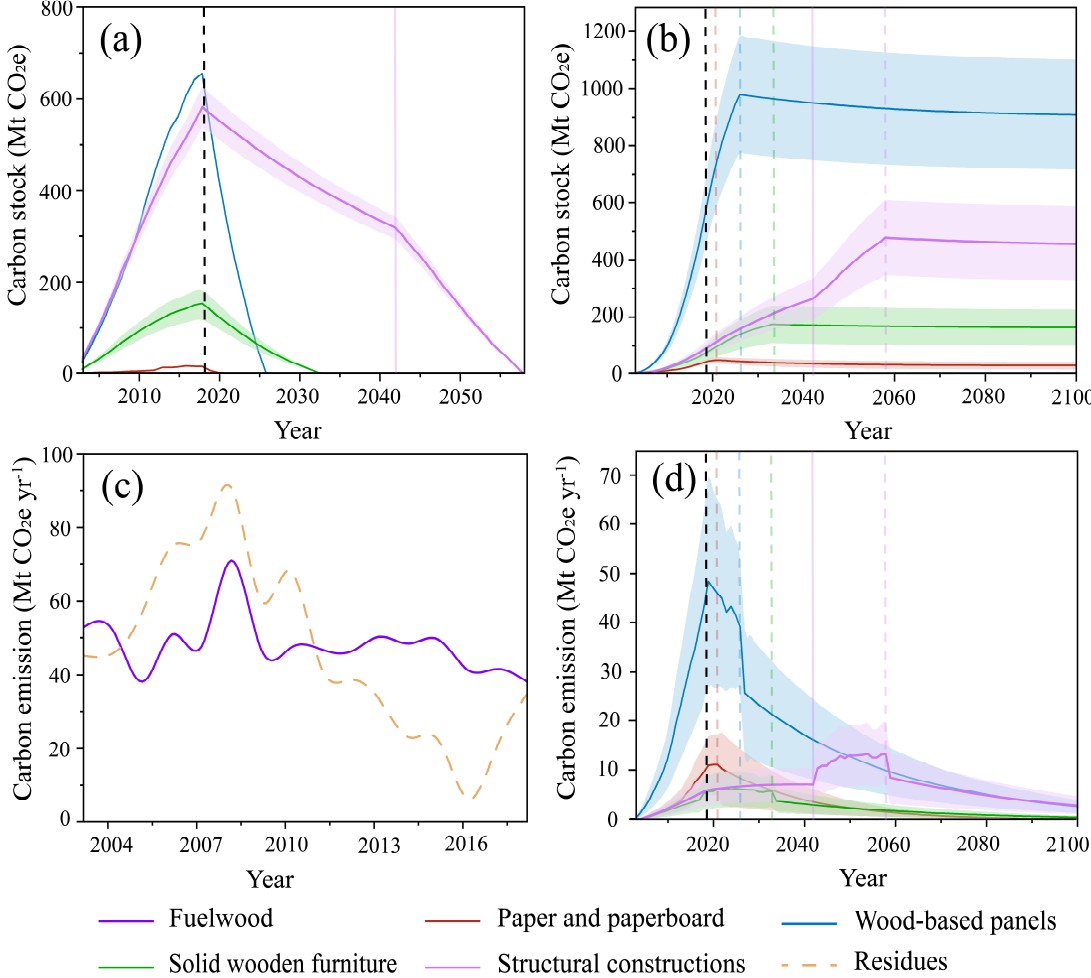

**Figure 9: The post-harvest carbon dynamics:** (a) the accumulated carbon stock of HWPs in-use; (b) the accumulated carbon stock in landfill; (c) annual carbon emissions of fuelwood and residues, since the burning of residues is not a definite fact, we represented it using dashed lines; (d) annual carbon emissions in landfill. The black dashed line indicates the year of 2018, when there is no new wood inflow

in the HWPs, the other dashed lines in (b) and (d) indicate the year when the corresponding products were all discarded, and the purple solid line indicates the year when the first construction wood reached its service life. Since paper and paperboard, wood-based panels, and solid wooden furniture have a service life of less than 16 years, carbon stock of emissions has no sharp changes before and after the year reached service life. The shaded area represents the variation range.

## 4. Discussion

### 4.1 Implications for simulating the carbon sink in China

Forest harvesting is one of the most important human activities determining the terrestrial ecosystem carbon budget (Liu et al., 2011; Pan et al., 2011). Forest harvesting largely decreases the leaf area index, increases litter biomass, and alters stand temperature and moisture, strongly impacting the structure and function of ecosystems (Nepstad et al., 1999; Liu et al., 2011; Jian et al., 2022). Harvesting causes damage to the forest canopy and soils, leading to temporary increases in litter carbon stock. It ultimately results in a net loss of soil carbon due to the damages inflicted on the forest canopy and soil. Following a disturbance, soil carbon loss can exceed carbon gain in above-ground biomass (Kowalski et al., 2004). Pennock and van Kessel (1997) found that soil carbon decreased by 5 to 20 t C ha$^{-1}$ over a 20-year period following clear-cutting, a significant loss compared to the carbon accumulated in a maturing forest's biomass (Pennock and Van Kessel, 1997).

Forest harvesting is a type of human activity that cannot be simulated by ecosystem modeling. Therefore, the development of regional and global datasets is necessary for quantifying the impacts of forest harvesting. The LUH2 dataset is an important source of data for indicating global forest harvesting and has been widely used to simulate the impacts of forest harvesting on the terrestrial carbon sink (Harper et al., 2018; Hurtt et al., 2020; Friedlingstein et al., 2022). However, our results showed that the LUH2 dataset underestimated harvested carbon by about 38.5% on average for China, compared with the statistical data (Fig. 5, Fig. S4). The LUH2 dataset used national wood volume harvest data from the Food and Agriculture Organization (FAO, 2020). Across all of China, the harvested carbon data from the FAO were approximately 40% higher compared to the statistical data averaged from 2003 to 2018, caused by the different statistical methodologies and data sources used in FAO and the China's statistical data. Nevertheless, in LUH2, the wood from agricultural expansion has been subtracted, the remaining national wood was then explicitly harvested (Hurtt et al., 2020). However, according to LUH2, the cropland area has increased by 41 million hectares since 1980 in China, which significantly deviates from

the actual situation (i.e., decreased by 14 million hectares) (Yu et al., 2022). Therefore, LUH2 has overestimated wood harvests due to agricultural expansion, leading underestimated of wood from forest harvesting. And after 2000, the spatial pattern of forest harvesting was constrained using the Landsat forest loss data (i.e., TCL) (Hansen et al., 2013) by verifying if the annualized gridded forest loss derived from the Landsat data matched the forest harvesting information in the LUH2 dataset. However, the TCL dataset only indicates clear-cutting and does not include selective logging. Our result showed that there was a large proportion of selective logging (Fig. 6), indicating that the LUH2 dataset largely underestimates the harvest area.

This study not only represents the temporal and spatial patterns of harvested carbon, but also provides the allocation of harvested carbon among the pools. Unlike harvested crop carbon, which will emit into the atmosphere at a faster rate, harvested forest carbon is stored in various HWPs pools, and emits back into the atmosphere with different lifetimes (Skog, 2008; IPCC, 2019a). As an extension of forest resources, the carbon dynamics of HWPs in use and after use have multiple impacts on national greenhouse gas (GHG) inventories (Johnston and Radeloff, 2019). Clarifying the proportion of post-harvest carbon allocated to different pools is the key to accurately assessing carbon dynamics. In addition, previous studies focused more on the carbon stored in HWPs (Stockmann et al., 2012; Matsumoto et al., 2022; Wei et al., 2023). However, the assessment of the global potential of HWPs as a carbon sink is subject to the balance between carbon inflows and outflows (Johnston and Radeloff, 2019). In particular, waste products emit non-$CO_2$ GHG such as $CH_4$, contributing to climate change (Cai et al., 2018). Moreover, the carbon emissions from harvested wood exhibit delayed and long-term effects. Tracking the long-term dynamics of carbon and assessing the future climate mitigation potential can provide a better foundation for GHG management in the forestry sector.

**4.2 Spatio-temporal changes of harvested carbon and allocation in China**

The spatial heterogeneity of forest harvesting is influenced by the distribution of forest resources in China (Fig. 4). Intensive harvesting in Southern China (e.g., Guangxi and Fujian) indicates a demand for specific tree species, such as *Eucalyptus* and *Cunninghamia lanceolata* (Yu et al., 2020). As the total harvesting increased, both clear-cutting and selective logging exhibited a general upward trend (Fig. 6), mirroring the dynamic equilibrium in the allocation of harvesting ways across China's forests. The rate of increase in clear-cutting significantly outpaced that of selective logging (Fig. 6). A growing demand

for wood is driving an upward in large-scale forest harvesting. Nevertheless, selective logging has remained the principal way of forest harvesting in China. For entire China, the pixels occurred selective logging is about 50 folds of that occurred clear-cutting (Fig. S5), and the mean harvested biomass in a pixel from selective logging is 8% of that from clear-cutting (Fig. S5, Fig. 6). The occurrence of such a small percentage of biomass removal at pixel level suggests the ability of the LEAF dataset to capture minor disturbances. Moreover, selective logging is a widely employed silvicultural practice that plays a central role in forest management worldwide (Liu et al., 2011). In the United States, the area covered by selective logging is approximately 61% of that occupied by clear-cutting (Masek et al., 2011). In Brazil, selective logging doubles the previous estimates of the total forest degraded by human activities (Asner et al., 2005). Selective logging is a more diffuse disturbance than forest clearance (Fisher et al., 2014) and is mostly invisible to satellites (Asner et al., 2005; Matricardi et al., 2010; Hethcoat et al., 2019). Despite many efforts to address the challenge of estimating selective logging using satellite data through image classification, it is still difficult to monitor low-intensity selective logging due to the coarse resolution of satellite imagery (Asner et al., 2005; Matricardi et al., 2010; Hethcoat et al., 2019). Objective spatially explicit reporting on selective logging is the basis for model-based assessment of the impact of forest harvesting on the carbon budget.

Forest harvesting in China continued to increase during the study period. However, the wood for fuel declined by nearly 23%, especially the harvesting for burning from farmers, contributing an average of nearly 70% of burned timber annually, decreased by nearly 35% (Fig. S6). This rapid decline in rural areas indicates that China is successful in the rural energy transition, benefiting the mitigation of air pollution and climate change (Chen et al., 2016; Tao et al., 2018). Production of wood and paper products in China was on the rise (Fig. 7), constituting a carbon pool that delays carbon release (Fig. 9). HWPs in use rarely emit carbon if there was no decay or combustion (e.g., building fire), being a stable carbon sink during their lifespan, especially wood used for constructions (Profft et al., 2009; Churkina et al., 2020). Although the $CO_2$ emitted in landfill is considered carbon neutral (Van Ewijk et al., 2020), $CH_4$ emissions from landfill are detrimental to climate due to their higher global warming potential (IPCC, 2023). Promoting an increase in carbon storage and a reduction of carbon emissions can be achieved by extending the lifespan and improving the recycling rate of HWPs (Brunet-Navarro et al., 2017). However, this approach may not be as effective for paper products, as they often have low or even negative recycling benefits (Van Ewijk et al., 2020). Therefore, recommendations in the wood sector prioritize

allocating harvested wood to long-lasting products (Fortin et al., 2012; Smyth et al., 2014) and products

with high recycling rates (Brunet-Navarro et al., 2016; Werner et al., 2010). Improving the efficiency of

wood harvesting and processing processes, as well as enhancing forest ecological management, is also

necessary for the reduction of carbon emissions.

### 4.3 Uncertainties of the LEAF dataset

Although the LEAF dataset demonstrated a good performance in capturing the spatial variability of

harvested carbon in China, several potential uncertainties exist. First, currently, the use of forest wood

was majorly extracted from the aboveground components. Typically, roots of logged trees will be

disposed by several ways, including decay stimulation, sprout regeneration, combustion, and the

production of small boards (Li et al., 2012). There are no detailed information regarding to roots due to

large differences over the regions and species. Therefore, this study did not include the root biomass in

the entire estimate, which may result in underestimation of HWPs and its subsequent emissions.

Second, the harvested carbon was found to be underestimated in several provinces over several

years. Among the estimates covering a span of 16 years across 31 provinces, approximately 28% of them

were underestimated, with the highest frequency of underestimation occurring in southern provinces (Fig.

10a). Compared to statistical values, the harvested carbon was underestimated by an average of 0.4-60.4%

from 2003 to 2018 across these investigated provinces (Fig. 10b). Anhui Province had the highest degree

of underestimation, with underestimation occurring in all years (Fig. 10). To avoid this underestimation,

we assigned the unidentified harvested carbon to the pixels where selective logging occurred, increasing

the harvested carbon of these pixels (Sect. 2.1, Eq. (6)), resulting in an estimation of more harvested

carbon than the actual AGB at several pixels. We examined the number of pixels where the harvested

carbon exceeded its AGB at the province level. The results showed that the number of pixels where

harvested carbon exceeded its AGB was less than 2% of the number of pixels where harvesting occurred

for each province annually (Fig, S7).

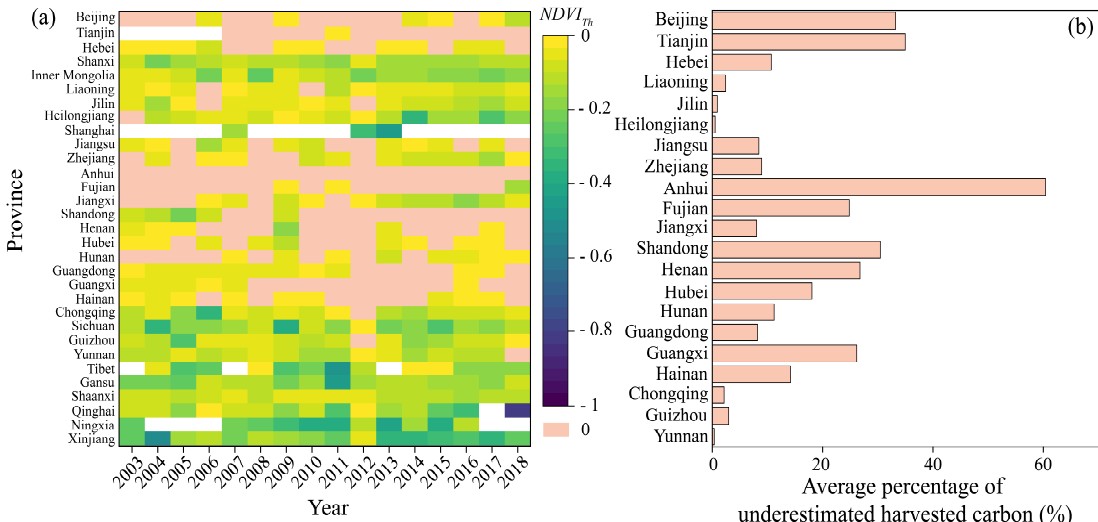

**Figure 10:** (a) Annual $NDVI_{diff}$ thresholds ($NDVI_{Th}$) for each province. An $NDVI_{Th}$ of 0 indicates that the province's harvested carbon was underestimated that year and a blank $NDVI_{Th}$ denotes that there was no harvesting in that province that year according to the statistics. (b) Average percentage of underestimated harvested carbon in the provinces where underestimation occurred from 2003 to 2018.

The underestimation of harvested carbon is closely related to the accuracy of identifying both selective logging and clear-cutting. (1) Wood output rate (R) is a key parameter for calculating the total wood harvest at provincial level. Its value varies depending on factors such as tree species, wood quality, and processing ways (Jiang et al., 2022). Even within the same province, R can vary significantly. However, due to the lack of R data at the sub-provincial level, this study utilized the provincial R values for 2009 obtained from the official website of National Bureau of Statistics (Table S2). Using provincial-level R for a single year overlooked the intra-provincial and inter-annual variations in R, potentially leading to bias in estimating the total wood harvest. (2) AGB is another important potential cause. After forest harvesting, trees regeneration would form new biomass. This study used the national forest inventory datasets to estimate AGB, which were conducted at a five-year interval. With this five-year period, the forest inventory can't include the increased biomass due to trees regeneration, which will lead to an underestimation of AGB. (3) The conversion of biomass storage to AGB (*Coef*) exhibited significant variations among provinces, possibly due to different dominant vegetation types and growing stock levels across provinces. The default biomass conversion and expansion factors (BCEF, equivalent to *Coef* in this study) provided by IPCC also confirmed the significant differences in *Coef* across vegetation types and growing stock levels (IPCC, 2006b). However, compared to IPCC's coarse regional classification, the provincial-level *Coef* we utilized in this study is more detailed. (4) The quality of selective logging identification largely depends on the quality and performance of NDVI. The accuracy

is largely impacted by low data quality in cloudy and rainy southern areas. In addition, NDVI is sensitive to green vegetation cover but not dense vegetation (Huete et al., 2002), and begins to saturate over dense vegetation with AGB values higher than 25 t C ha$^{-1}$ (Chang et al., 2023), contributing to the underestimation of AGB. Overlooking the continued growth after harvesting within a pixel and the regional-scale climate impacts led to an underestimation of $NDVI_{diff}$. Consequently, the harvested carbon may be underestimated at regional scales, particularly in provinces with substantial forest coverage (e.g., Guangxi). (5) The estimation of clear-cutting was primarily limited by the performance of the TCL dataset generated by Hansen et al. (2013). In subtropical and temperate regions, the producer's accuracy and user's accuracy of TCL are both approximately 80%, indicating that the dataset can identify tree cover loss with relatively high accuracy. However, there is still a 20% level of uncertainty (Hansen et al., 2013), and further efforts are needed to improve the identification of clear-cutting.

Third, we compared our estimates of carbon stocks of HWPs with previous studies. Zhang et al. (2019) estimated the carbon stock of HWPs to be 1.7 Gt $CO_2$e for the period of 2003-2016, which is 1.3 times higher than our estimates for the same period (Fig. 8b). Zhang et al. (2019) used statistical data of HWPs provided by FAO, which has a significant disparity to China's official figures (Fig. 6). Zhang et al. (2018) estimated carbon stocks of HWPs from 1950 to 2015 using China's official data, but which also included the imported HWPs. As the China's wood imports are considerable, the estimates of Zhang et al. (2018) for 2003-2015 is nearly 1.8 times of our estimates. This study aims to quantity the contribution of harvested wood to national $CO_2$ emission in China; therefore, we excluded the imported wood according to the IPCC standard. Moreover, these previous estimates depended on default factors recommend by IPCC inventory method to calculate $CO_2$ emissions and stocks in HWPs, and which showed large differences with specific factors in China used by this study. In addition, this study also provides estimates of stocks and $CO_2$ emissions dynamics, which is quite important for understanding its long-term contributions. Nevertheless, information on the end products of harvested wood (i.e., bathroom tissue, tableware, types of furniture, columns, etc.) is unavailable (Profft et al., 2009), leading to rough estimates of wood destination and product service life. In this study, all residual wood was assumed to be burned as fuelwood would induce an overestimation of carbon emissions. Meanwhile, this study ignored recycled wood products, which may extend the carbon storage in products (Brunet-Navarro et al., 2017), causing an underestimation of carbon stock in products in use. More efforts are needed to track the production and end use of the harvested wood, as they play a key role in the estimation of the effects

on national GHG inventories (White et al., 2005; Johnston and Radeloff, 2019).

## 5. Data availability

The 30 m × 30 m Long-term harvEst and Allocation of Forest Biomass (LEAF) dataset is available

at https://doi.org/10.6084/m9.figshare.23641164.v2.(Wang et al., 2023).

The file format of the product is GeoTIFF with the spatial reference of WGS84 (EPSG:4326). Each GeoTIFF file represents annual harvested forest carbon (unit: g C m$^{-2}$yr$^{-1}$), the absolute harvested carbon per pixel is obtained by multiplying by the corresponding pixel area.

## 6. Conclusion

This study produced a Long-term harvEst and Allocation of Forest Biomass (LEAF) dataset, which provides spatial information on forest harvesting at a resolution of 30 m. The validation results demonstrated the accuracy and reliability of the LEAF dataset in capturing the spatial variation of harvested carbon. From 2003 to 2018, harvested carbon showed an increased trend. Additionally, our dataset showed that selective logging resulted in more than 80% of the total harvested carbon. The carbon

taken away from forest harvesting was allocated to six wood pools, with the direct combustion of fuelwood as the primary source of carbon emissions after harvesting. However, it is important to highlight that the carbon stored in wooden products has the potential for long-term retention. In summary, the development of the LEAF dataset enhances our understanding of the spatial patterns of forest harvesting and post-harvest carbon dynamics in China. The LEAF dataset generated by this study is an important

data source for estimating the carbon budget of forest ecosystems in China, which can also provide essential insights for sustainable forest management and climate change mitigation efforts.

## Author contributions

WY and DW designed the research, performed the analysis, and wrote the paper; PR, and XX performed the analysis; LF, ZQ, and XC edited and revised the manuscript.

**Competing interests**

The authors declare that they have no conflict of interests.

**Financial support**

This research was supported by the key project of the National Natural Science Foundation of China (42141020).

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
