# Peer review of "National forest carbon harvesting and allocation dataset for the period 2003 to 2018"

_Earth System Science Data, 2023_

## Author Comment (AC1)

**Response to Referee #1**

**This study generated nation-wide harvested carbon data through synthesizing the remote sensing data, statistical data and empirical equations/parameters, and further estimated the carbon stock in harvested wood. The dataset can help provide effective and accurate data and parameters for estimating carbon flux due to forest disturbance and harvesting.**

Thanks for your positive feedbacks. We deeply appreciate your time and efforts in reviewing the manuscript, and we have significantly revised the manuscript according to your comments.

**The major problems in this manuscript include:**

**1. The methods are not detailed and sound enough, which result in the less confidence for this work.**

Thanks for your comment. We understand the referee's concerns regarding the level of detail and reliability of our methods. In the revised manuscript, we provided more comprehensive and detailed information of method. Please refer to responses #6, #8, #11, #13, #16, and #18.

**2. Lack of a quantification of uncertainty.**

Yes, it is a good idea to provide the quantification uncertainty, and we have quantified the uncertainties for decayed carbon in the revised manuscript according to your comment. Please refer to response #21 for details.

**3. Some of the results seem unreasonable.**

We understand your concerns about the results. We followed your comments to improve the method and ensure the results reliable. Please refer to responses #19 and #20.

**Other specific comments/suggestions are listed below.**

**4. There are a lot of English writing issues. I won't point them here. The authors should carefully revise the English writing sentence by sentence.**

Thanks for your comment. We thoroughly checked and improved English usages of the revised manuscript and asked a professional company (www.InsiderOfScience.com) to polish the English.

**5. Some terminologies are not conformed throughout the manuscript. Please check through and make them consistent.**

Thank you for such detailed comments and reminders. We have checked and revised the terminologies throughout the text for consistency.

**6. Line 107-140: This approach for calculating the selective harvesting is not valid enough since NDVI values varied greatly among years due to climate change and other disturbance events. The CCDC, VCT, LandTrendr and other similar algorithms are more professional and proved methods to detect partial forest disturbance. You can consider using one of those algorithms to replace your work.**

We've studied the algorithms you recommended and analyzed their feasibility of identifying selective logging.

CCDC (Continuous Change Detection and Classification) utilizes all available Landsat imagery to build time series models for each pixel. Changes are identified by comparing the difference between modeled pixel value and observed value against a predefined threshold. The algorithm mentions that more subtle changes (e.g., selective logging) can be captured by using smaller thresholds. However, the performance and accuracy in detecting selective logging has not been verified. And the current algorithm does not explicitly provide a parameter adjustment scheme for performing selective logging identification (Zhu et al., 2014).

VCT (Vegetation Change Tracker) first calculates a composite forest z-score index (IFZ) for each pixel to determine if it was forest. Then, the derived forest index images are stacked to form an IFZ time series for each pixel, which is then analyzed to detect and track forest changes (Huang et al., 2009). For a detected disturbance, its disturbance magnitude is calculated as the difference between the IFZ value at the disturbance year and the mean IFZ value within the concerned time series. The disturbance magnitude can serve as an indicator of whether a disturbance is a major (e.g., clear-cutting) or minor disturbance (e.g., selective logging). However, the validation showed that the VCT was poor at capturing minor disturbances such as selective logging (Huang et al., 2009; Thomas et al., 2011).

LandTrendr (Landsat-based Detection of Trends in Disturbance and Recovery) divides time series spectral trajectories into linear sequence segments. According to the direction of spectral change associated with any given sequence segment in the trajectory, the segment was given the label of

disturbance, recovery, or stabilization. LandTrendr can effectively capture sharp disturbances, such as clear-cutting. Although disturbance intensity can be reflected by the magnitude of change in spectral values, LandTrendr does not specify how to effectively detect low intensity disturbance, such as selective logging (Kennedy et al., 2010).

In summary, the above algorithms have great potential to identify large forest disturbance, however, there are no studies confirm their abilities for detecting selective logging effectively. We genuinely value your insightful suggestions and appreciate your understanding that implementing these algorithms to reach our intended outcomes is quite time-consuming. We will continue to maintain and update our dataset, considering integrating and comparing these specialized algorithms in future research to enhance the credibility and accuracy of our dataset.

Our method assumed selective logging activities would result in a larger decrease of satellite-based vegetation index (e.g., NDVI) compared to forest disturbances like droughts, heatwaves, and insect outbreaks (Yuan et al., 2014; Gao et al., 2019). Based on the different responses of NDVI to logging and other disturbances, separating logging from other disturbances and climate change can be achieved by setting a threshold for NDVI decline. Specifically, logging is identified only when the NDVI decrease surpasses this set threshold. We elaborated on these separation techniques in the second assumption of our methodology (refer to lines 128-140 in the revised manuscript). The validations against the surveyed forest harvesting at 133 cities and counties indicated a good performance of our method (Fig. 3, Fig. S2 in the revised manuscript).

**7. Line 152-155: In fact, paper belongs to the wood product pool (pool 3). "wood fuel" can be renamed to "fuelwood".**

Yes, we have reclassified and renamed the post-harvest wood pools according to this and other comments, including fuelwood pool, paper and paperboard pool, wood-based panels pool, furniture pool, constructions pool, and residues pool (refer to response #8 for detail).

**8. (a) Line 153-159: To more accurate tracking the carbon fates in HWPs, varied time-scale HWP pools should be further divided. For example, divide HWP into 2 (such as paper), 5 (e.g., decorative uses), 20 (e.g., furniture), 50 (e.g., buildings) half-life HWP pools.**

Yes, the classification that you recommended is consistent with that of statistical data, therefore

we divided HWPs into paper and paperboard, wood-based panels (including decorative uses), furniture, and constructions (buildings) based on their lifetime. We have revised the description of HWPs classification and recalculated the emission accordingly in the revised manuscript.

"The harvested wood was allocated into six wood pools (Fig. 2). (1) Fuelwood pool, where the wood is burned as fuel, resulting in immediate carbon emissions through combustion; (2) paper and paperboard pool, including household paper, printing paper, packaging paper, etc.; (3) wood-based panels pool, including fiberboard and particle board made from wood residues (such as barks, branches, sawdust) or small stems bonded with adhesives, are commonly used as decorative panels for various applications like wall cladding and ceiling finishes; (4) furniture pool, referring to wooden household items such as tables, chairs, wood beds, etc.; (5) constructions pool, referring to the structural components used to support buildings, such as beams, columns, and trusses; and (6) residues pool, including leaves, killed understory vegetation, and unutilized wood residues, which are typically left on the logging site or treated as fuel (Lippke et al., 2011; Stockmann et al., 2012), and were assumed as fuel in this study. The wood pools of (2), (3), (4), and (5) belong to HWPs pool, where the carbon will remain stored until the products are either retired from use or reach the end of their service life (Table S4) and are consequently discarded. Subsequently, these discarded products are then sent directly to landfill, where they decompose.

[Figure]

**Figure 2:** The allocation of post-harvest wood to the six wood pools.

In this study, the volume ($m^3$) of fuelwood, pulpwood (wood for paper and paperboard), woodbased panels, and the sum volume (m$^3$) of wood for furniture and constructions can be obtained directly from the China Forestry Statistical Yearbook. The constructions and furniture pools were allocated as the percentage of 74.61±6.24% and 25.39±6.24% from their sum, respectively, according to China Timber and Wood Products Distribution Industry Yearbook. The wood in pools (1) to (5) was converted into carbon, with conversion factors listed in Table S3 (IPCC, 2019b). Then, the carbon entering the residues pool can be calculated by subtracting the carbon in pools (1) to (5) from the total harvested carbon (Sect. 2.3.2)."

**Table S3:** The carbon conversion factor of roundwood and wood-based panels.

| Wood categories | Carbon conversion factor (t C m$^{-3}$) |
|---|---|
| Roundwood | 0.229 |
| Wood-based panels | 0.269 |

**(b) In addition, a landfill carbon pool is needed to separate since the decay of the landfill is significantly different from the regular HWPs. The carbon decay from landfill should be separately simulated.**

Yes, the carbon decay from landfill has been separately simulated, and the corresponding figures and results in the manuscript have been revised accordingly. (Lines 365-400; Fig. 9)

**9. Line 158: In my memory, the half-life of paper in IPCC report is 2 years (4 years' lifespan)? Please double check and confirm.**

**Line 158:In this study, the lifespan of paper products and wood products were assumed to be 5 and 100 years (IPCC, 2019a), respectively, and we calculated the delayed emissions to 2100.**

Yes, this study used 2 years for the half-life of paper as the reviewer mentioned. According to IPCC (2019b), the half-life is a function of the country-specific service life of particular HWPs (HL=*service life* × *ln* (*2*)). According to the comment #11, we used more specific parameters, and revised new parameter of paper lifespan instead of 5 years (Table S4; c response #11).

**10. Line 158: it is not reasonable and accurate to assign 100 years' span (50 years for half-life) for all HWPs. As I suggest above, the HWPs should be separated more pools to accurately track the carbon dynamic.**

Yes, according to your advice, we have used new detailed classification of the HWPs (refer to

response #8), and we searched and used new parameters for lifespan (refer to response #11).

**11. Line 167-222: Most of the parameter values in this study are from the IPCC default values. Actually, more specific parameter values are available for China. Please retrieve more published papers from Chinese scholars related to forest harvesting or disturbance or harvested wood.**

As your suggestion, we retrieved more specific parameter values for China from Chinese statistical websites, national standards, published papers, questionnaires and experimental analyses from other scholars. Including (1) the service life, which was also used to calculate half-life (*HL*) of HWP categories in-use (Table S4), (2) key parameters of $CH_4$ generation in landfills (mean values of various landfills types were used in this study) (Table S5), (3) decay constant and the proportion of discarded organic carbon ($DOC_f$) that can be decomposed under anaerobic conditions for each type of waste in landfills (Table S6). The descriptions in section 2.2.2 of the methodology were revised accordingly.

**Table S4:** Service life of HWPs categories in-use.

| HWP categories | Service life (year) |
|---|---|
| Paper and paperboard | 3[a] |
| Wood-based panels | 8[a] |
| Furniture | 15[a] |
| Constructions | 40[b] |

[a] Wang et al., 2017; [b] IPCC, 2019b.

**Table S5:** Key parameters of $CH_4$ generation in landfills.

| Parameters | Value ($\pm$SD) |
|---|---|
| $CH_4$ oxidation factor ($OX_T$) | 0.176 ($\pm$0.06) |
| $CH_4$ correction factor ($f_{ar}$) | 0.28 ($\pm$0.15) |
| $CH_4$ recovery rate ($R_T$) (%) | 12.06 ($\pm$9.95) |
| Volume fraction of $CH_4$ | 0.5 ($\pm$0.1) |

Values were derived from Cai et al. (2018).

**Table S6:** Decay constant (*k*) and the proportion of discarded organic carbon ($DOC_f$) that can be decomposed under anaerobic conditions for various waste types in landfills.

| Type of waste | *k* | $DOC_f$ |
|---|---|---|
| Paper | 0.05 ($\pm$0.01)[b] | 0.5 ($\pm$0.2)[a] |
| Wood-based panels | 0.03 ($\pm$0.01)[b] | 0.1 ($\pm$0.05)[c] |
| Furniture | 0.028 ($\pm$0.01)[b] | 0.105 ($\pm$0.025)[c] |
| Construction | 0.025 ($\pm$0.01)[b] | 0.105 ($\pm$0.025)[c] |

[a] IPCC, 2019a; [b] Cai et al., 2018; [c] Grann, 2015.

**12. Line 185: the SWDS overlaps with the landfill carbon. I suggest discarding the concept of SWDS and using "Landfill" to replace.**

Thanks for your suggestion, we have replaced "SWDS" by "landfill".

**13. Method: how to allocate the harvested tree organs (leaf/root/stem) to the four harvested carbon pools? All the harvested biomass is allocated into HWPs? The fractions for these organs vary greatly among tree species and ages. The allocation methods are needed to elaborate in the method.**

After forest harvesting, leaves are typically left on the logging site, becoming logging residues that are allocated to the residues pool (National standard: LY/T 3135-2019). The stems (i.e., wood) are used as fuel or to produce wood products that will be allocated to the fuelwood pool and HWPs pools (refer to response #8). In this study, due to the unclear treatment of logging roots in China, only harvested aboveground biomass was counted, and roots were not included. We added several sentences to discuss the uncertainties.

"Currently, the use of forest wood was majorly extracted from the aboveground components. Typically, roots of logged trees will be disposed by several ways, including decay stimulation, sprout regeneration, combustion, and the production of small boards (Li et al., 2012). There are no detailed information regarding to roots due to large differences over the regions and species. Therefore, this study did not include the root biomass in the entire estimates which may result in underestimation of HWPs and its subsequent emissions." (Lines 501-508)

We have elaborated on the allocation methods in more detail in our methodology (refer to comment #8).

**14. Method: I did not see how the pixel level AGB is obtained or calculated (only mentioned the provincial AGB data). This is important to ensure the data quality.**

Equation (1) (Line 105) and (2) (Line 112) outlined the methodology for computing the pixel level AGB. Briefly, provincial-level AGB derived from forest inventory dataset was allocated to pixel-level based on the proportion of each forest pixel's NDVI to the sum of NDVI values for all forest pixels at a given province. (Refer to lines 103-116 for details)

**15. Method: If I understand right, the inventory AGB is used to calculate the harvested carbon;**

**however, this static AGB data fails to track the forest regrowth after harvesting. This will result in some uncertainties, so a short discussion is needed to mention this limitation.**

Thanks for your valuable comment. We have added a short discussion for the uncertainty of the AGB.

"After forest harvesting, trees regeneration would form new biomass. This study used the national forest inventory datasets to estimate AGB, which were conducted at a five-year interval. With this five-year period, the forest inventory can't include the increased biomass due to trees regeneration, which will lead to an underestimation of AGB." (Line 527-530 in the revised manuscript)

**16. Method: there are too many parameters and their values. A or multiple Table(s) should be used to list all parameters, values and sources.**

We have added three tables to list all parameter values and sources (Table S2, Tables S4~S6) (refer to response #11).

**17. Line 218: the GWPs for CH$_4$ in the new IPCC report should be used rather than the IPCC 2007.**

Thanks for your suggestion. We have adopted the 100-year GWPs for CH$_4$ provided in the latest IPCC report (the Sixth Assessment Report), with a value of 28 (IPCC, 2023), replacing the value provided by IPCC 2007.

**18. Line 281: The harvested carbon here includes which carbon pools? All four carbon pools? Or just harvested wood biomass?**
**Line 281: Accuracy evaluation of the LEAF dataset generated from this study showed good performance in indicating spatial variations of harvested carbon in China.**

In our revised version, the harvested carbon includes all six carbon pools, we have listed all harvested carbon and its allocation in Figure 2 (refer to response #8).

**19. Line 304: 80% harvested carbon is from selective harvesting. This percentage is too high. In my mind, the main forest harvesting mode is clear cut, especially for the commercial forests. Any additional proofs to provide?**

Thanks for your deep thoughts. It is a very important issue to quantify the percentage of clearcutting and selective logging. However, there are no statistics to provide the valid information. We appreciate your understanding that we can't provide additional proofs. Yes, you are right at some regions, for example southern China with large area of commercial forests, both clear-cutting and selective logging are prevalent, and clear-cutting may be a major harvest type (e.g., Jiangxi, Guangdong, and Yunnan) (Fig. 4c). However, it may reasonable that a significant portion of forest logging in most provinces in China is indeed dominated by selective logging, based on the following reasons. (1) According to Forest Law of the People's Republic of China starting from 2000 (http://www.npc.gov.cn/zgrdw/npc/flsyywd/xingzheng/node_2169.htm), clear-cutting has been be strictly controlled. Clear-cutting is only applicable to overmature single-layer forests and uneven-aged forests with small fraction of young- and middle-aged trees. In general, the area for clear-cutting should not exceed 5 hectares at once. However, most forests are currently in the young- and middle-aged categories in China (Zhang et al., 2017), not applicable for clear-cutting. (2) Shelterbelts and forests designated for specific purposes, such as national defense forests, seedling forests, environmental protection forests, and scenic forests, are permitted only for nurturing and selective logging for regeneration purposes. (3) At 2015, China has completely ceased commercial logging in key state-owned forest areas, and by 2017 the injunction has been extended to all natural forests in China, which implies clear-cutting is strictly prohibited. Only selective logging for forest nurturing can be conducted. In summary, we believe it is reasonable for high percentage of selective logging in China benefiting from national forestry protection and management policies.

**20. Figure 8: The annual variations in harvested carbon pools look unreasonable, for example, why the total carbon stock (all carbon pools add together) is still very high at 2100? Based on the decay equations and half-life span (50 half-life for HWPs), most of the harvested carbon should be released at 2100 (about 80 years from 2019-2100). Need to double check the calculation equations and their parameters, especially for the carbon decay equations.**

Thank you for deep thoughts. In the figure, carbon stock is presented as an accumulated quantity rather than an annual flux., and we revised the figure caption (Lines 405-411 or refer to response #46). The half-life for HWPs in use refers to the rate at which HWPs are phased out from use, and stored carbon in the HWPs will not immediately released but instead ends up in landfills. In addition, not all stored carbon in landfills can be decomposed and return to atmosphere. Actually, the decomposed

fraction varied from 0.1 to 0.5, depending on type of organic waste materials (Table S6), and the stored carbon in landfills was decomposed gradually. Therefore, several years later, a significant portion of the carbon deposited in landfills remains stored there.

**21. Results: Uncertainty ranges should be quantified when present the research results (harvested carbon and decayed carbon) since there are a lot uncertainties associated to parameters and their values. Namely, each result should provide a variation range (±standard deviation). Single parameter values are provided in the equations, but actually these values could vary in a range.**

Thanks for your constructive comment. For decayed carbon, we provided a variation range for each result (Fig. 9) based on the specific parameter values we retrieved (Tables S4~S6). While it difficult to quantify the uncertainties for harvested carbon. First, provincial statistics for biomass and harvest were a defined value, with no variation range. Second, pixel-scale biomass and harvested wood were calculated based on satellite-based vegetation index which is also difficulty to quantify the uncertainties. We appreciate your understanding, and we have discussed all the potential uncertainties in depth in the discussion section (Section 4.3, Lines 501-548 in the revised manuscript).

**22. Discussion: the estimated HWPs amounts in this study should be compared with previous studies. As I know, there are several studies published by the Chinese scholars also provided these data.**

Thanks for your thoughtful comment. We have retrieved several previous studies on carbon stock and emissions from HWPs in China, and made comparisons with our result.

"We compared our estimates of carbon stocks of HWPs with previous studies. Zhang et al. (2019) estimated the carbon stock of HWPs to be 1.7 Gt $CO_2$e for the period of 2003-2016, which is 1.3 times higher than our estimates for the same period (Fig. 8b). Zhang et al., (2019) used statistical data of HWPs provided by FAO, which has a significant disparity to China's official figures (Fig. 6). Zhang et al., (2018) estimated carbon stocks of HWPs from 1950 to 2015 using China's official data, but which also included the imported HWPs. As the China's wood imports are considerable, the estimates of Zhang et al. (2018) for 2003-2015 is nearly 1.8 times of our estimates. This study aims to quantity the contribution of harvested wood to national $CO_2$ emission in China; therefore, we excluded the imported

wood according to the IPCC standard. Moreover, these previous estimates depended on default factors recommend by IPCC inventory method to calculate $CO_2$ emission and stock in HWPs, and which showed large differences with specific factors in China used by this study. In addition, this study also provides estimates of stock and $CO_2$ emission dynamics, which is quite important for understanding its long-term contributions." (Line 552-564 in the revised manuscript)

**23. The uploaded data link on figshare does not work, please check the problem.**

Thanks for your reminder, we double checked the link and it worked. We are going to check it in the near future to ensure it work.

**Response to Referee #2**

**With intensified climate change, international organizations and national governments have highlighted the significant role of forests in mitigating climate change. This research attempts to establish a long-term harvest and allocation of forest biomass dataset, which provides guidelines for China's carbon budget assessment. The analysis of forest carbon allocation would attract more attention in future research. However, I still have some questions, which I'm hoping authors can respond to:**

Thanks for your positive feedback. We deeply appreciate your hard work for reviewing the manuscript. We revised the manuscript based on the comments.

**24. The dataset is the central component of this study, but its name doesn't appear to accurately describe its features. What are the benefits of this dataset, furthermore? The dataset's significance is not properly represented.**

The name of dataset is Long-term harvEst and Allocation of Forest Biomass (LEAF) dataset, which is one component of the Terrestrial Ecosystem Disturbance (TED) dataset, referred as TED-LEAF (Lines 85-86). First, Terrestrial Ecosystem Disturbance (TED) dataset indicates to represent the spatial information of primary anthropogenic disturbances in forest, cropland, and grassland ecosystems, aimed at providing foundations and references for China's terrestrial carbon budgets. Second, the name of Long-term harvEst and Allocation of Forest Biomass (LEAF) dataset includes two important components of this dataset: location of harvested wood and its allocation to six wood pools, and we try to express these two core features in the name. If there are any potential names proposed by the reviewer, we would like to consider to use.

As a critical terrestrial ecosystem type, forests play a pivotal role in carbon cycle. Forest harvesting is the most significant anthropogenic disturbance of forests. However, the impacts of forest harvesting on the terrestrial carbon cycle has not been estimated yet due to a lack of available data on harvested carbon. The Land-Use Harmonization 2 dataset offers harvested carbon data, but its resolution is coarse for analyzing regional or local forest ecosystem dynamics, especially within China's context (Hurtt et al., 2020). Furthermore, carbon harvested from forests doesn't release immediately but decomposes over time due to various wood uses. Despite being a major wood consumer, China lacks a comprehensive estimate of carbon dynamics within its wood products. As

stated by referee #1 and referee #3, our dataset can help provide effective and accurate data and parameters for estimating carbon flux due to forest disturbance and harvesting, as well as a better understanding of the scale and impacts of forestry.

**25. It is necessary to introduce the design of the dataset establishment in further detail at the beginning of the approach. Otherwise, readers cannot catch your key points.**

Thanks for your suggestion. We have further refined our method description by including more detailed information (refer to response #26 and #30).

**26. I have other issues: what's the difference between forest loss and selective logging? It is best to describe new concepts as they emerge in your research for the first time and demonstrate how they relate to earlier concepts.**

Thanks for your comment. Forest loss generally refers to a reduction in forest area caused by natural factors (e.g., wildfires, pests) or human activities (e.g., logging, agricultural expansion). In the tree cover loss dataset generated by Hansen et al., (2013), forest loss is defined as the stand replacement disturbance or the complete removal of tree cover at a 30 m scale (Hansen et al., 2013). In this study, it's defined as clear-cutting, which is one type of logging ways. Selective logging represents another logging way. Overall, clear-cutting and selective logging are two logging way within human-induced forest loss activities.

We have added a description of clear-cutting and selective logging to the text.

"In this study, clear-cutting is the harvesting of an entire stand at once on a scale of 30m × 30m, while selective logging is the harvesting of a portion of the stand within that area that is suitable and should be harvested." (Lines 95-97 in the revised manuscript)

**27. For the same reason, what does the term "potential selective logging areas" mean, and why?**

"Potential selective logging areas" refer to the regions or locations within a forested area where selective logging activities might have occurred. In this specific context, the term is used to describe areas where the NDVI has decreased between two adjacent years. Since this decrease is not solely attributed to selective logging but also includes other factors such as droughts, heatwaves, and insect outbreaks, these areas are considered to be potential sites for selective logging. By distinguishing selective logging from other disturbances, we identified the areas where actual selective logging took

place from these areas. (Refer to Lines 128-140 in the revised manuscript for details)

**28. The research contains many acronyms. It is preferable to create a table so that they can be seen more clearly.**

Thanks for your suggestion. We have provided a table to explain the acronyms (Table S1).

**Table S1:** Full names and abbreviations of terminologies.

| Full name | Acronyms | Full name | Acronyms |
|---|---|---|---|
| Long-term harvest and Allocation of Forest Biomass | LEAF | Harvested Wood Products | HWPs |
| Land-Use Harmonization 2 | LUH2 | Tree Cover Loss | TCL |
| First-order decay | FOD | Above-ground biomass | AGB |
| National Forest Inventory | NFI | Harvested carbon from clear-cutting | $HC_C$ |
| Normalized Difference Vegetation Index | NDVI | Harvested carbon from selective logging | $HC_S$ |
| Statistical harvested carbon | SHC | Estimated service life | ESL |
| Discarded organic carbon | DOC | Greenhouse gas | GHG |

**29. Line 110: Why do you assume that? Is there a foundation?**

**Line 110: This approach relied on two fundamental assumptions. First, we assumed that NDVI values decreased resulting from selective logging. Second, we assumed that the reductions in NDVI values resulting from selective logging would be more significant compared to decreases caused by other factors such as droughts, heat waves, ice storms, and insect outbreaks (Yuan et al., 2014).**

The Normalized Difference Vegetation Index (NDVI) can effectively monitor vegetation density and greenness, with higher NDVI values indicating denser and healthier vegetation. For a pixel, several biomasses would be removed due to logging, and several times is needed to regenerate. Consequently, vegetation appears sparser in the year following logging, leading to a decrease in NDVI compared to the previous year. Additionally, disturbances like droughts, heatwaves, and insect outbreaks can reduce NDVI by impacting vegetation health, as well. However, the reduction in NDVI due to these disturbances is less significant compared to the complete biomass removal from logging. Hence, areas showing a significant decrease in NDVI can be prioritized for logging, and statistical logging data can help determine the threshold of NDVI decrease. Based on these foundations, Yuan et al. (2014) accurately identified afforestation locations in China by detecting the changes of NDVI between two

adjacent years (Yuan et al., 2014). Therefore, we expected to use a similar approach to identify logging locations based on these foundations.

**30. It is important to consider the basis upon which your approach or theory is based. This is the weakness of this study.**

Thanks for your deep thoughts. In this study, we identified the location and scale of forest harvesting and tracked post-harvest carbon dynamics. Forest harvesting is done in two ways: clear-cutting and selective logging. For clear-cutting, the tree cover loss (TCL) dataset produced by Hansen et al., (2013) has provided valuable information of location. Based on TCL, the location of clear-cutting can be effectively recognized. For selective logging, the method was based on the principle of multitemporal satellite-based vegetation index analysis and detected the changes of the index between two adjacent years. As we responded to comment #29, NDVI, a commonly used vegetation index, is closely related to the vegetation canopy greenness and is widely used for vegetation monitoring and disaster assessment. By analyzing changes in NDVI in two adjacent years, areas of reduced forest cover can be effectively identified, and final harvest areas can be determined by combining statistical harvest data (refer to response #29 for details).

For post-harvest carbon dynamics, the IPCC has provided a solid theoretical foundation. The IPCC first recognized harvested wood products (HWPs) in their Revised 1996 IPCC Guidelines for National Greenhouse Gas Inventories (IPCC, 1997), but expressed a default assumption that "all carbon biomass harvested is oxidized in the removal/harvest year". Then, in their 2006 Guidelines, the IPCC revised their treatment of HWPs, emissions from the HWPs pool were assumed to follow a decay function (IPCC, 2006a). Subsequently, IPCC continued revising and enhancing the estimation of carbon dynamics within HWPs, encompassing the disposal of HWPs in-use and the decay emissions from discarded HWPs in landfills. Currently, the theoretical framework and methodologies regarding HWPs have been progressively refined. Leveraging China-specific activity data based on these methodologies, we estimated the carbon dynamics of Chinese HWPs, aiding the assessment of carbon benefits in Chinese forestry.

We agree with your point that basis upon of the approach or theory is important. We added a description of the theoretical foundations of the methodology of this study at the beginning of the approach.

"This study aimed to generate a Long-term harvEst and Allocation of Forest Biomass (LEAF) dataset, which is a component of the Terrestrial Ecosystem Disturbance (TED) dataset, referred to as TED-LEAF. The LEAF dataset includes the location and scale of forest harvesting and the estimates of post-harvest carbon dynamics. The identification of forest harvesting is based on the detection of changes in multi-temporal vegetation indices. Combined with statistical harvest data, the forest harvesting and other disturbances causing such changes can be separated. Utilizing the classification of HWPs provided by statistical data, we estimated the delayed carbon emissions by 2100 from HWPs based on IPCC methodologies with China-specific activity data." (Lines 85-92 in the revised manuscript)

**31. Generally, linear regression is used to represent the average level. However, the scale of dataset established in this study is 30 m * 30m, therefore the test based on linear regression cannot be a reliable way to verify the accuracy of LEAF dataset.**

Thanks for your comment. In this study, the province-level statistical harvested carbon was used to determine the threshold for identifying selective logging. It is difficult to obtain harvesting information at pixel scale, this study used city and county-level surveyed harvested carbon to examine the dataset performance. We have tried to collect the city and county-level surveyed harvested carbon information as much as possible from the official website of each provincial forestry bureau (refer to section 2.3.3 in the revised manuscript). Finally, there were 133 records nationwide available from 2006 to 2018 and 14 provinces (Fig. S1). By aggregating the estimated harvested carbon from pixel level to the city and county level and comparing it to the collected surveyed harvested carbon, we found that our data captured the spatial distribution of harvested carbon well at the national scale (Fig. 3), across provinces (Fig. S2), and across years (Fig. S3). Furthermore, the surveyed data we collected comes from administrative units of different levels, can represent various spatial scales. Therefore, we believe this kind of comparison can be used to validate the spatial distribution of the forest harvesting map.

**32. The data scales before and following the section on "carbon flows between HWPs" appear to be inconsistent, and the part appears to have little link with the preceding sections. It is not creative to just calculate the current state of carbon flows.**

Forest harvesting removes carbon from forest ecosystems but does not return it to the atmosphere

immediately, where it is stored in HWPs for years to decades. The turnover rate of its return to the atmosphere depends on the end use of the wood. The HWPs is an extension of forest resources and play a crucial role in forest's carbon cycle.

A high-resolution forest harvesting dataset is fundamental for accurately estimating China's national carbon budget and serves as a cornerstone for ecosystem modeling. As well as offers vital insights for sustainable forest management. However, allocating harvested carbon at a pixel scale lacks significance and feasibility. Due to the unavailability of data on commercial wood and product flows among provinces, delayed carbon release from harvested wood is only calculated to the national scale.

The understanding of carbon dynamics within harvested wood in China remains limited. In ORCHIDEE and HN2017 models, carbon release assumptions were relatively simplified: ORCHIDEE presumed an even distribution over product residence times (10 and 100 years), while HN2017 assumed exponential decay across the same residence times (Yue et al., 2020). In bookkeeping models, HWPs were allocated to three product pools with turnover times of 1, 10, and 100 years (Hansis et al., 2015). However, this study highlights that the bookkeeping model's allocation proportions significantly differ from the actual wood utilization in China. Therefore, there's a critical need to further categorize China's harvested carbon, quantifying its distribution among various wood pools, and closely monitoring its carbon dynamics. Such detailed analysis is expected to provide more precise and reliable data and parameters, essential for estimating carbon flux resulting from forest disturbance and harvesting in China.

**33. Also, if I understand correctly, the core of the article's dataset construction is "the TCL dataset produced by Hansen et al. (2013)", and then used Chinese coefficients to calculate forest carbon. How significant it is? Additionally, the coefficients are applied at the provincial level, which is significantly less precise than carbon sink observations from remote sensing.**

The TCL dataset produced by Hansen et al. (2013) is one of the basic datasets for our dataset construction, not the core. The TCL dataset has provided the spatial location of complete tree cover removal at the 30 m scale, which provided valuable information for the identification of clear-cutting locations in this study. However, the TCL dataset does not include the magnitude of the harvested biomass, which need to be calculated using other datasets, e.g., grid forests biomass data. We calculated biomass at the pixel level rather than simply using the China coefficients (Lines 102-116 in the revised

manuscript). Moreover, the statistical wood production data show a much larger portion of the wood harvest comes from partial tree removal (selective logging). Therefore, the aim of this study is to provide the spatial location and magnitude of all forest harvesting with a spatial resolution of 30 m and quantify the post-harvest carbon dynamics.

Indeed, remote sensing can provide more precise observations. However, there is currently no available long-term biomass dataset at a 30 m scale in China. In this study, combining NDVI and provincial-level biomass data, we calculated the pixel-level biomass. By combining the expansive spatial coverage of remote sensing with the authoritative statistical data, we accurately mapped out the spatial distribution of forest harvesting.

**Response to Referee #3**

**Wood harvesting is a significant land use activity that reduces forest biomass and affects carbon budget. This paper developed a spatial dataset of harvested carbon pools in China by downscaling province-level wood harvest statistics using satellite data. This dataset also includes the allocation of wood products over time. The idea of merging two data sources is very constructive, and I believe such a dataset of wood harvests is critical for understanding the scale and impact of forestry. While this dataset could have a very strong scientific contribution, I have several concerns, mainly regarding the methodology, which require clarification and careful examination by the authors.**

Thanks for your positive comments and taking time out of your busy schedule to review our manuscript. We revised the manuscript based on the comments.

**Major comments:**

**The data generation are separated into two parts: 1) Harvested carbon, and 2) Decay of harvested carbon pools.**

**Part 1.**

**34. The way that estimates selective logging seems problematic (Equation 5). The quantity of selective logging is determined by the percentage of NDVI reduction from the previous year. How do the authors separate the regrowth after harvest and selective logging within the same grid cell? $NDVI_{diff}$ can be the net result of regrowth and loss due to logging. In the algorithm, the actual selective logging will be considered as disturbance (undetected) when $NDVI_{diff}$ is small. There is a risk of underestimating selective logging in areas with higher forest regrowth, where $NDVI_{diff}$ is smaller.**

In this study, we did not consider the regrowth after harvest, and we have discussed this uncertainty in the manuscript (Lines 540-541 in the revised manuscript).

We agree with you that $NDVI_{diff}$ should be the net result of regrowth and selective logging within the same grid cell. However, our identifying is conducted at the annual scale. The tree regrowth is a gradually process, and the NDVI changes resulting from regeneration is quite slight if considering interannual variations of climate variables. Furthermore, NDVI represents indirectly leaf biomass, and

the allocation of the net primary productivity to leaves is less than 30% averaged forests of various stand age, and which is lower at the old stand age (Xia et al., 2019). On contrary, logging is a dramatic disturbance which may lead to large decrease of NDVI. Therefore, the uncertainty arising from overlooking the regrowth of forests post-harvest is relatively minor. Your comment is excellent, and in future studies, this issue should be considered if data becomes available, to enhance the accuracy of forest harvest estimations.

**35. The way that calculates carbon loss due to clear-cutting seems straightforward. However, the parameter Coef (L95-100, Table S1) requires some data quality checks and some discussion of potential data uncertainties.**

**For example, there is a huge difference in Coef between Qinghai and Tibet despite similar ecozones (even though forests are rare over there). Coef is equivalent to wood basic density x carbon fraction of dry wood x biomass expansion factor. Assuming wood basic density and carbon fraction at 0.5, biomass expansion factor will be around 2.8 to get a Coef at 0.73 in Qinghai, which is extremely high. This raises a question of whether this Coef method is valid in areas with very low AGB values. Also, there is a typo in Table S1: Qinhai-> Qinghai.**

Thanks for your valuable comment. First, the ecozones where the Tibetan and Qinghai forests are located are different (Xia et al., 2023). The forests in Tibet are primarily concentrated in its southeastern mountains, with most of these areas belonging to the subtropical zone, featuring a balanced of broadleaf and coniferous forests. In Qinghai, the forests lie in the high-altitude climate zone and are predominantly coniferous.

Second, the 9th National Forest Inventory indicated that the growing stock level in Qinghai and Ningxia is less than 20 $m^3$ $ha^{-1}$. According to the default biomass conversion and expansion factors (BCEF, equivalent to Coef in this study) provided by the IPCC, the values for Qinghai also seem high (IPCC, 2006b). As the IPCC lacks a specific category for high-altitude climate zones and considering that the areas where Qinghai's forests are located are closer to temperate zones, we use data from the temperate zone as a reference (Table R1).

**Table R1:** Default biomass conversion and expansion factors (BECF) (t C $m^{-3}$)

| Forest type | Growing stock level ($m^3$ $ha^{-1}$) | | | | |
| --- | --- | --- | --- | --- | --- |
| | < 20 | 21-40 | 41-100 | 100-200 | >200 |

| | | | | | |
|---|---|---|---|---|---|
| Hardwoods | 1.5 (0.4-0.225) | 0.85 (0.4-1.3) | 0.7 (0.35-0.95) | 0.525 (0.3-0.7) | 0.4 (0.275-0.55) |
| Pines | 0.9 (0.3-1.2) | 0.5 (0.325-0.75) | 0.375 (0.3-0.5) | 0.35 (0.2-0.5) | 0.35 (0.2-0.5) |
| Other conifers | 1.5 (0.35-2) | 0.7 (0.25-1.25) | 0.5 (0.25-0.7) | 0.375 (0.2-0.6) | 0.35 (0.175-0.45) |

Third, in this study, we utilized provincial forest area (ha), forest volume (m$^3$), and aboveground biomass carbon density (t C ha$^{-1}$) data from the National Forest Inventory to calculate the *Coef*, which extrapolates harvested wood volume to aboveground biomass carbon. Since aboveground biomass encompasses both the canopy layer and understory vegetation (shrubs, herbs, etc.) (National standard: LY/T 2988-2018), the *Coef* expansion encompasses not only the wood components—branches, bark, leaves—but also the biomass of the understory vegetation. This might explain why the *Coef* in Qinghai is relatively high, as its sparse canopy allows more growth for understory vegetation. Consequently, despite lower forest volume levels, the higher carbon density is observed.

We have revised "Qinhai" to "Qinghai". Meanwhile, we added a discussion of the uncertainties of the AGB calculation.

"The underestimation of harvested carbon is closely related to the accuracy of identifying both selective logging and clear-cutting. AGB is an important potential cause. After forest harvesting, trees regeneration would form new biomass. This study used the national forest inventory datasets to estimate AGB, which were conducted at a five-year interval. With this five-year period, the forest inventory can't include the increased biomass due to trees regeneration, which will lead to an underestimation of AGB. The conversion of biomass storage to AGB (*Coef*) exhibited significant variations among provinces, possibly due to different dominant vegetation types and growing stock levels across provinces. The default biomass conversion and expansion factors (BCEF, equivalent to *Coef* in this study) provided by IPCC also confirmed the significant differences in *Coef* across vegetation types and growing stock levels (IPCC, 2006b). However, compared to IPCC's coarse regional classification, the provincial-level *Coef* we utilized in this study is more detailed." (Lines 527-536 in the revised manuscript)

**36. Can the authors explain why FAO data is substantially higher (40-50% more) than province level data (Figure 5)? What are the potential systematic biases between the two statistics?**

In this study, the province level data was provided by the China's Forestry Statistical Yearbook. The following reasons contribute to the bias between FAO and province level data.

First, the statistical methods used by FAO and China's Forestry Statistical Yearbook are different. The harvested wood volume provided by FAO includes all trees harvested and removed from forests or other logging sites (e.g., agricultural protection forest). It includes stem wood, roots, and branches from harvesting and removal of trees killed or damaged by natural causes (e.g., fire, insects and diseases) (FAO, 2020). However, the data recorded in China's Forestry Statistical Yearbook solely accounts for the quantity of wood harvested from forests and transported to designated storage yards or allocated points, meeting the national timber standards upon measurement (i.e., wood output) (National Forestry Administration, 2000). Based on the provincial wood output rates (i.e., the ratio of wood output to the actual logging volume) provided by China's timber production plan from the National Bureau of Statistics (Table S1), we calculated the actual logging volume (refer to section 2.3.2 in the revised manuscript). In addition, the roots of logged trees were not considered in this study.

Second, the data source for FAO and China's Forestry Statistical Yearbook are different. The harvested wood volume from FAO is primarily based on data obtained from annual forestry product questionnaires responded by various countries or from official figures. However, the domestic harvested wood for China is mainly estimated based on questionnaire results, not from official figures (https://www.fao.org/faostat/en/#data/FO). Unfortunately, FAO failed to specified the methods and processes used in the estimation, limiting further analysis of the reasons for the bias between FAO and China's Forestry Statistical Yearbook.

**37. I recalled that LUH2 also used FAO data, so why is there a huge discrepancy between FAO and LUH2. Please explain by further elaborating the steps of calculation/aggregation of data.**

Yes, the LUH2 dataset used national wood volume harvest data from the FAO. The following reasons contribute to the huge discrepancy between FAO and LUH2.

First, in LUH2, wood harvests due to agricultural expansion were subtracted from the total harvest. The remaining harvested wood was then explicitly allocated to forest pixels (Hurtt et al., 2020). However, the land use change in LUH2 misestimated the changes in cropland area in China. According to LUH2, the cropland area has increased by 41 million hectares since 1980 in China, which significantly deviates from the actual situation. In fact, China's cropland has decreased by 14 million hectares since 1980 (Yu et al., 2022). Therefore, a significant amount of harvested wood was incorrectly attributed to cropland expansion, subtracting from the total harvest.

Second, the spatial patterns of forest transitions, particularly those related to wood harvesting, were constrained by the Landsat-based gridded forest loss observations from Hansen et al. (2013). However, the forest loss dataset only indicates clear-cutting and does not include selective logging at 30 m × 30 m scale (Hansen et al., 2013; Hurtt et al., 2020), indicating that the LUH2 dataset largely underestimates the harvest area.

Therefore, there is there a huge discrepancy between FAO and LUH2. In fact, we have discussed this in Section 4.1. To provide a clearer explanation for the readers, we further elaborated on this discussion as follows:

"The LUH2 dataset used national wood volume harvest data from the Food and Agriculture Organization (FAO, 2020). Across all of China, the harvested carbon data from the FAO were approximately 40% higher compared to the statistical data averaged from 2003 to 2018, caused by the different statistical methodologies and data sources used in FAO and the China's statistical data. Nevertheless, in LUH2, the wood from agricultural expansion has been subtracted, the remaining national wood was then explicitly harvested (Hurtt et al., 2020). However, according to LUH2, the cropland area has increased by 41 million hectares since 1980 in China, which significantly deviates from the actual situation (i.e., decreased by 14 million hectares) (Yu et al., 2022). Therefore, LUH2 has overestimated wood harvests due to agricultural expansion, leading underestimated of wood from forest harvesting." (Lines 433-442 in the revised manuscript)

**Part 2.**

**38. The paper uses confusing terminologies and too many acronyms, which makes it hard to follow.**

Sorry for the confusion. We have checked and simplified the terminologies throughout the text for consistency. And for the acronyms, we provided a table for their explanation (Table S1) (refer to response #28).

**39. Harvested wood products were allocated into wood fuel, paper, and wood products (L150, L230). How can the term "wood products" be used to refer both to the total sum, and to a sub-category? Suggest use longlived wood products.**

Thanks for your constructive comments. We have reclassified and renamed the harvested wood

products based on lifespans, including paper and paperboard, wood-based panel, furniture, and constructions (refer to response #8).

**40. Also, are landfill and solid waste disposal sites (SWDS) referring to the same thing?**

Yes, the SWDS overlaps with the landfill, we have replaced "SWDS" by "landfill".

**41. The definition of lumber that includes paper and wood fuel (2.3.2) is unconventional. Lumber is typically understood as wood used for building and furniture.**

Sorry for the confusion. We have revised "lumber for paper" to "pulpwood". For the terminological consistency and readability, we use "wood" to describe all primary or processed wood products. For example, we revised the wood categories as:

"Commercial wood was further divided into roundwood, pulpwood, and fuelwood. Non-commercial wood included the volume of wood logged ($F_{log}$) by farmers for burning and for their personal consumption." (Refer to Lines 264-281 for more details).

**42. Allocation method - it is not clear to me how the authors allocate the harvested wood into the four pools (Section 2.2, L150-155). For example, the residual pool includes wood used as wood fuel burned for energy, independent from the existing wood fuel category. How did the authors separate statistical data for wood fuel into dedicated wood fuel and residual wood being burned?**

In practical terms, the residual wood typically has two fates: left on the logging site or used as fuel. The usage for fuel is independent of the fuelwood provided by statistical data. In the previous version, we assumed all residual wood was burned as fuel. Figure 8a also depict fuelwood and residual wood separately, since the burning of all residual wood is not a definite fact, we represented it using dashed lines.

In the revised version, we have provided a more detailed and accurate categorization of harvested carbon, along with a more detailed description of the allocation methodology. (Refer to response #8, or Section 2.2 in the revised manuscript)

**43. How did the authors treat the harvesting slashes such as branches, killed understory vegetation, and roots in these pools? I suspect they are not accounted for and I am not sure whether the statistical data would provide this information. How did the authors treat bark**

**during the processing? Is it allocated to the wood fuel, residual, or not accounted for?**

As we responded in comment #36, the harvested biomass refers to all aboveground biomass, not contain roots. Branches and bark serve as crucial raw materials for fiberboard and particleboard, allocated to the wood-based panels pool. The killed understory vegetation and leaves are usually left on site to enhance soil fertility or treated as fuel, allocated to the residues pool. Statistical data does not directly provide information on components such as branches, bark, or killed understory vegetation. We extensively detail the calculations and allocations of these components in the manuscript (Refer to response #8, or Section 2.2 in the revised manuscript).

**44. The spatial distribution of the decay harvested carbon after harvesting (Figure 7). Did the authors assign the subsequential changes in the carbon pools to their original harvest locations (grid cell and province level)?**

It is nearly impossible to obtain the details about wood usage at the pixel scale, our calculations focused on the allocation of harvested wood at provincial level. Figure 7 (Figure 8 in the revised manuscript) was created to illustrate the usage of harvested wood (such as fuelwood and pulpwood) across provinces. This illustration represents the allocation of harvested wood within each province and does not signify that these woods will necessarily be processed or used within the same province.

**45. Considering that commercial wood is often transported and processed elsewhere, how is the spatial flow accounted for?**

We did not account for the spatial flow of harvested wood, and we appreciate your understanding it beyond the scope of this study. As you stated, the commercial wood is often transported and processed elsewhere, and end products like paper and furniture are usually consumed elsewhere, as well. Due to the unavailability of data on commercial wood and product flows, delayed carbon release from harvested wood is only calculated to the national scale.

**46. Figure 8b. (a) The linear increase in wood harvests at the beginning reflects the actual wood harvests from statistics during 2003-2018. Since you don't have harvesting data since 2018, the post-2018 time series are solely decay pools (not reflecting the real-world situations). These two periods represent different meanings and should perhaps be presented separately for clarity.**

Thanks for your valuable suggestions. We have revised Figure 8 (Figure 9 in the revised

manuscript).

In the carbon emission time series post-harvest, there are three critical time points: (1) 2018, when no new wood is input; (2) the year when the first batch of HWPs reaches the end of their service life; (3) the year when the final batch of HWPs reaches the end of their service life and is completely phased out. For each category of HWPs, we have labeled these three time points in the Figure.

[Figure]

**Figure 9:** The post-harvest carbon dynamics: (a) the accumulated carbon stock of HWPs in-use; (b) the accumulated carbon stock in landfill; (c) annual carbon emissions of fuelwood and residues, since the burning of residues is not a definite fact, we represented it using dashed lines; (d) annual carbon emissions in landfill. The black dashed line indicates the year of 2018, when there is no new wood inflow in the HWPs, the other dashed lines in (b) and (d) indicate the year when the corresponding products were all discarded, and the purple solid line indicates the year when the first construction wood reached its service life. Since paper and paperboard, wood-based panels, and furniture have a service life of less than 16 years, carbon stock of emissions has no sharp changes before and after the year reached service life. The shaded area represents the variation range.

**(b) It is also hard to compare 8a and 8b as the units are different.**

Yes, we harmonized the units of both carbon stocks and carbon emissions as $CO_2e$.

**47. Lifespan are 5, 100 years for paper and longlived products (IPCC, 2019a) in L155, while k for paper products and wood products were used as 0.347 and 0.023 (L180), respectively,**

**according to the IPCC default values (IPCC, 2014). Therefore, half-life will be ln(2)/0.347 and ln(2)/0.023, rather than 2.5 and 50 years. Which set of values is ultimately used?**

Sorry for the confusion. Here, the half-life of a product in-use refers to the number of years it takes for the product to be retired due to functional, technological, or economic obsolescence, resulting in a fifty percent reduction in its usage. Lifespan refers to the service life of a product which is known to be expected under a particular (reference) set of in-use conditions, depending on the workmanship and maintenance of the product. The half-life of a product is not half of its lifespan, but a function of the country-specific service life of particular HWPs (HL=*service life* × *ln* (*2*)) (IPCC, 2019b). According to the IPCC default values, half-life for paper and wood products are 2 and 30 years, respectively, corresponding to $k$ of 0.347 and 0.023. For instance, considering paper products put into use in 2003, they would be retired within decay constant of 0.347 annually before 2008. By 2008, all remaining products inflowed in 2003 would be entirely phased out due to reaching the end of their service life.

In our revised version, we have performed a more detailed classification of the HWPs and retrieved more specific parameter values for the estimation of decay carbon (refer to response #8, #9 and #11, or Section 2.2).

**Minor comments:**

**48. Figure 3. It is hard to compare a and b as the scales in the legends are so different. It also suggests that 30m resolution is less capable in capturing harvesting compared to the 0.1 degree. Response:** Thank you for pointing that out. In areas where harvesting is less frequent, pictures produced at a 30 m resolution may show visual gaps as the pictures are non-editable. To enhance readers' comprehension of the spatial patterns of forest harvesting, we offered three views: a map rendered at a 30 m resolution, a zoomed-in view at the same resolution, and a map that upscales the 30 m data to a resolution of 0.1°. The dataset we produced and uploaded is still 30 m, although the 0.1° map visually captures the spatial pattern of forest harvesting better than the 30 m, in practical applications, such as evaluation at the regional scale and as the base data for modeling, the accuracy of the 30 m data is higher.

We added a description of subgraph (b) to the caption of Figure 3 (Figure 4 in the revised version) as follows:

"**Figure 4:** Map of forest harvested carbon for China in 2016 at (a) 30 m and (b) 0.1°resolution and the zoomed-in view of the example areas of (a) (a1, a2, and a3), the map at 0.1° was derived from a 30 m data upscaling, and (c) shows the harvested carbon from clear-cutting and selective logging by province in 2016."

**References**

Cai, B., Lou, Z., Wang, J., Geng, Y., Sarkis, J., Liu, J., and Gao, Q.: CH4 mitigation potentials from China landfills and related environmental co-benefits, Sci. Adv., 4, eaar8400, https://doi.org/10.1126/sciadv.aar8400, 2018.

China News: China's State Forestry Administration: China has completely ceased commercial logging in natural forests. 2017. https://www.chinanews.com.cn/. (In Chinese)

FAO: Forestry data, FAOSTAT Database, Food and Agriculture Organization of the United Nations, Rome, Italy, Available at: https://www.fao.org/faostat/en/#data/FO, 2020, last access: 25 June 2023.

Gao, Y., Quevedo, A., Szantoi, Z., Skutsch, M.: Monitoring forest disturbance using time-series MODIS NDVI in Michoacan, Mexico, Geocarto International, 36, 1768–1784, 2019.

Grann, B.: A review of carbon loss from wood products in anaerobic landfills., ACLCA. Vancouver, BC, Canada, https://doi.org/10.13140/RG.2.2.34605.59365, 2015.

Hansen, M. C., Potapov, P. V., Moore, R., Hancher, M., Turubanova, S. A., Tyukavina, A., Thau, D., Stehman, S. V., Goetz, S. J., Loveland, T. R., Kommareddy, A., Egorov, A., Chini, L., Justice, C. O., and Townshend, J. R. G.: High-resolution global maps of 21st-century forest cover change, Science, 342, 850–853, https://doi.org/10.1126/science.1244693, 2013.

Hansis, E., Davis, S. J. & Pongratz, J.: Relevance of methodological choices for accounting of land use change carbon fluxes. Global Biogeochem. Cycles, 29, 1230–1246, 2015.

Huang, C. Q., Goward, S. N., Schleeweis, K., Thomas, N. E., Masek, J. G., Zhu, Z. L.: Dynamics of national forests assessed using the Landsat record: Case studies in eastern United States, Remote Sens. Environ., 113, 1430–1442, https://doi.org/10.1016/j.rse.2008.06.016, 2009.

Hurtt, G. C., Chini, L., Sahajpal, R., Frolking, S., Bodirsky, B. L., Calvin, K., Doelman, J. C., Fisk, J., Fujimori, S., Klein Goldewijk, K., Hasegawa, T., Havlik, P., Heinimann, A., Humpenöder, F., Jungclaus, J., Kaplan, J. O., Kennedy, J., Krisztin, T., Lawrence, D., Lawrence, P., Ma, L., Mertz, O., Pongratz, J., Popp, A., Poulter, B., Riahi, K., Shevliakova, E., Stehfest, E., Thornton, P., Tubiello, F. N., Van Vuuren, D. P., and Zhang, X.: Harmonization of global land use change and management for the period 850–2100 (LUH2) for CMIP6, Geosci. Model Dev., 13, 5425–5464, https://doi.org/10.5194/gmd-13-5425-2020, 2020.

IPCC: Refinement to the 2006 IPCC Guidelines for National Greenhouse Gas Inventories. Volume 5: Waste. Chapter 3: Solid Waste Disposal, Available at: https://www.ipcc-nggip.iges.or.jp/public/2019rf/pdf/5_Volume5/19R_V5_3_Ch03_SWDS.pdf, 2019a, last access: 25 June 2023.

IPCC: Refinement to the 2006 IPCC Guidelines for National Greenhouse Gas Inventories. Volume 5: Waste. Volume 4: Agriculture, Forestry and Other Land Use. Chapter 12: Harvested Wood Products, Available at: https://www.ipcc-nggip.iges.or.jp/public/2019rf/pdf/4_Volume4/19R_V4_Ch12_HarvestedWoodProducts.pdf, 2019b, last access: 27 December 2023.

IPCC: Climate Change 2021 – The Physical Science Basis: Working Group I Contribution to the Sixth

Assessment Report of the Intergovernmental Panel on Climate Change, 1st ed., Cambridge University Press, https://doi.org/10.1017/9781009157896, 2023.

IPCC: Revised 1996 Intergovernmental Panel on Climate Change Guidelines for National Greenhouse Gas Inventories. Paras, France, 1997. https://www.ipcc-nggip.iges.or.jp/public/gl/invs6d.html.

IPCC: 2006 IPCC Guidelines for National Greenhouse Gas Inventories. Volume 4: Agriculture, Forestry and Other Land Use. Chapter 12: Harvested Wood Products, Available at: https://www.ipcc-nggip.iges.or.jp/support/Primer_2006GLs.pdf, 2006a, last access: 25 June 2023.

IPCC: 2006 IPCC Guidelines for National Greenhouse Gas Inventories. Volume 4: Agriculture, Forestry and Other Land Use. Chapter 4: Forest Land, Available at: https://www.ipcc-nggip.iges.or.jp/public/2006gl/pdf/4_Volume4/V4_04_Ch4_Forest_Land.pdf, 2006b.

Kennedy, R. E., Yang, Z., and Cohen, W. B.: Detecting trends in forest disturbance and recovery using yearly Landsat time series: 1. LandTrendr — Temporal segmentation algorithms, Remote Sens. Environ., 114, 2897–2910, https://doi.org/10.1016/j.rse.2010.07.008, 2010.

Li, F., He, B., and Yang, Z.: Progress of research on tree felling root treatment and application technology., Pract. For. Technol., 9–11, https://doi.org/10.13456/j.cnki.lykt.2012.08.017, 2012.

Lippke, B., Oneil, E., Harrison, R., Skog, K., Gustavsson, L., and Sathre, R.: Life cycle impacts of forest management and wood utilization on carbon mitigation: knowns and unknowns, Carbon Manag., 2, 303–333, https://doi.org/10.4155/cmt.11.24, 2011.

LY/T 2988-2018: Guideline on carbon stock accounting in forest ecosystem. Beijing: Standard Press of China, 2018.

LY/T 3135-2019: Wood residues. Beijing: Standard Press of China, 2019.

National Forestry Administration: The definition of China forestry statistic indicators. China Forestry Press, Beijing, China. pp: 117-137. 2000.

Stockmann, K. D., Anderson, N. M., Skog, K. E., Healey, S. P., Loeffler, D. R., Jones, G., and Morrison, J. F.: Estimates of carbon stored in harvested wood products from the United States forest service northern region, 1906-2010, Carbon Balance Manag., 7, 1, https://doi.org/10.1186/1750-0680-7-1, 2012.

Thomas, N. E., Huang, C. Q., Goward, S.N., Powell, S., Rishmawi, K., Schleeweis, K., Hinds, A.: Validation of North American Forest Disturbance dynamics derived from Landsat time series stacks, Remote Sens. Environ., 115, 19–32, https://doi.org/10.1016/j.rse.2010.07.009, 2011.

Wang, H.Y., Zuo, X., Wang, D. L., and Bi, Y. Y.: The estimation of forest residue resources in China, J. Cent. South Univ. Forestry Technol., 37(02), 29-38, https://doi.org/10.14067/j.cnki.1673-923x.2017.02.006, 2017.

Xia, J., Yuan, W., Lienert, S., Joos, F., Ciais, P., Viovy, N., Wang, Y., Wang, X., Zhang, H., Chen, Y., and Tian, X.: Global Patterns in Net Primary Production Allocation Regulated by Environmental Conditions and Forest Stand Age: A Model‐Data Comparison, J. Geophys. Res. Biogeosciences, 124, 2039–2059, https://doi.org/10.1029/2018JG004777, 2019.

Xia, X., Xia, J., Chen, X., Fan, L., Liu, S., Qin, Y., Qin, Z., Xiao, X., Xu, W., Yue, C., Yue, X., and Yuan, W.: Reconstructing long‑term forest cover in China by fusing national forest inventory and 20 land use and land cover data sets, J. Geophys. Res. Biogeosciences, 128, e2022JG007101, https://doi.org/10.1029/2022JG007101, 2023.

Yu, Z., Ciais, P., Piao, S., Houghton, R. A., Lu, C., Tian, H., Agathokleous, E., Kattel, G. R., Sitch, S., Goll, D., Yue, X., Walker, A., Friedlingstein, P., Jain, A. K., Liu, S., and Zhou, G.: Forest expansion dominates China's land carbon sink since 1980, Nat. Commun., 13, 5374, https://doi.org/10.1038/s41467-022-32961-2, 2022.

Yuan, W., Li, X., Liang, S., Cui, X., Dong, W., Liu, S., Xia, J., Chen, Y., Liu, D., and Zhu, W.: Characterization of locations and extents of afforestation from the Grain for Green Project in China, Remote Sens. Lett., 5, 221–229, https://doi.org/10.1080/2150704X.2014.894655, 2014.

Yue, C., Ciais, P., Houghton, R. A., and Nassikas, A. A.: Contribution of land use to the interannual variability of the land carbon cycle, Nat. Commun., 11, 3170, https://doi.org/10.1038/s41467-020-16953-8, 2020.

Zhang, L., Sun, Y., Song, T., and Xu, J.: Harvested wood products as a carbon sink in China, 1900–2016, Int. J. Environ. Res. Public. Health, 16, 445, https://doi.org/10.3390/ijerph16030445, 2019.

Zhang, X., Yang, H., and Chen, J.: Life-cycle carbon budget of China's harvested wood products in 1900–2015, For. Policy Econ., 92, 181–192, https://doi.org/10.1016/j.forpol.2018.05.005, 2018.

Zhang, Y., Yao, Y. T., Wang X. H., Liu Y. W., and Piao, S. L.: Mapping spatial distribution of forest age in China. Earth Space Sci. 4, 108-116. https://doi.org/10.1002/2016EA000177, 2017.

Zhu, Z., and Woodcock, C. E.: Continuous change detection and classification of land cover using all available Landsat data, Remote Sens. Environ., 144, 152–171, http://dx.doi.org/10.1016/j.rse.2014.01.011, 2014.

---

## Author Response (AR2)

MS NO.: essd-2023-309

MS Type: Data description paper

Title: National forest carbon harvesting and allocation dataset for the period 2003 to 2018

Dear editor and referee,

We appreciate the valuable opportunity to further revise our manuscript "National forest carbon harvesting and allocation dataset for the period 2003 to 2018" (MS No.: essd-2023-309) for possible publication in ESSD.

The remaining issues pointed out by referee #3 have been carefully addressed. Specifically, we further clarified the allocation of harvested carbon to wood pools. The corresponding text in the Methods have been carefully revised.

Please find attached the point-by-point responses to the comments of the referee. Please note that the comments from the referee are in **bold** followed by our responses in regular text. The changes in our manuscript are underlined with red.

Thank you for your consideration.

Sincerely,

Daju Wang, Wenping Yuan, on behalf of all co-authors
Email: yuanwp3@mial.sysu.edu.cn

**Response to Referee #3**

**I notice that the authors redo the wood product pools (Fig.2).**

**1. First, Fig.2 is still extremely confusing to me. The relationship is unclear between the statistical harvesting data (in section 2.3.2, commercial and noncommercial) and the allocation of the product pools (in section 2.2). Fig.2 mixes the raw wood materials with the products.**

Sorry for the confusion. We revised Fig. 2, and it now clearly shows how the statistical harvesting data been allocated to six wood pools.

[Figure]

**Figure 2:** The allocation of post-harvest wood to six wood pools. The green boxes indicate the variables are available from the China Forestry and Grassland Statistics Yearbook. R is wood output rate of commercial wood, and *Coef* (t C m$^{-3}$) is the coefficient that converts harvested wood (m$^3$) to harvested carbon (t C) (i.e., biomass carbon of trunks, branches, leaves and understory vegetation).

We revised the corresponding description in section 2.2 and section 2.3.2.

**Section 2.2:**

"The harvested wood was allocated into six wood pools (Fig. 2). (1) Fuelwood pool, the sum of commercial fuelwood and non-commercial fuelwood (i.e., farmers' fuelwood), where the wood is burned as fuel, resulting in immediate carbon emissions through combustion; (2) paper and paperboard pool, including household paper, printing paper, packaging paper, etc.; (3) wood-based panels pool,

including plywood, fiberboard, particle board, and other wood-based panels, made from roundwood, wood residues (such as barks, branches, sawdust) or small stems bonded with adhesives, are commonly used as decorative panels for various applications like wall cladding and ceiling finishes; (4) solid wooden furniture pool, referring to solid wooden household items such as tables, chairs, wood beds, etc.; (5) structural constructions pool, referring to the structural components used to support buildings, such as beams, columns, and trusses; and (6) residues pool, including leaves, killed understory vegetation, and unutilized wood residues, which are typically left on the logging site or treated as fuel (Lippke et al., 2011; Stockmann et al., 2012), and were assumed as fuel in this study."

**Section 2.3.2:**

"The annual provincial wood output ($m^3$), extracted from the China Forestry and Grassland Statistical Yearbook, was categorized into two main types: commercial wood and non-commercial wood (Fig. 2). Commercial wood included fuelwood and roundwood, and roundwood was further categorized into pulpwood, wood for plywood, and other roundwood (e.g., roundwood for directly use, internally processed roundwood, etc.). Non-commercial wood included farmers' fuelwood (i.e., the volume of wood logged by farmers for burning) and farmers' self-use wood (i.e., the volume of wood logged by farmers for their personal consumption). Non-commercial wood refers to the actual logged volume, which can be totally used by farmers. Commercial wood refers to the wood output volume of peeled wood that meets the national wood standards, not the total volume of wood logged from the forest (National Forestry Administration, 2000). After wood logged, preliminary processing (such as peeling, sawing, etc.) is carried out and some unusable or poor-quality wood is eliminated. Therefore, the commercial wood output is generally less than the actual logged wood (National Forestry Administration, 2000). Based on the provincial wood output rate (R, i.e., the ratio of commercial wood output to wood logged) provided by China's timber production plan from the National Bureau of Statistics (Table S2), we calculated the actual annual wood logged for each province as:

$$W_{log} = \frac{W_{output}}{R} \tag{20}$$

where $W_{log}$ is the actual annual wood logged for producing commercial wood, $W_{output}$ is the volume of commercial wood output. Then, the total harvested carbon (i.e., the SHC in Sec. 2.1) for a given province was calculated as:

$$SHC = (W_{log} + F_{log}) \times Coef \tag{21}$$

where $F_{log}$ is the volume of non-commercial wood, and *Coef* (t C m$^{-3}$) is the coefficient that converts harvested wood (m$^3$) to harvested carbon (t C) (i.e., biomass carbon of trunks, branches, leaves and understory vegetation) (Table S2)."

**2. (a) Wood-based panels refer to different "board" products made from wood chips, veneers, sawdust, strands, or fibers. Most of the source materials are from manufacturing sawn wood and it is rare to include barks and branches as wood-based panels. Therefore, one can consider wood-based panels as part of roundwood.**

Thanks for such detailed comment. Plywood is indeed derived from roundwood, other wood-based panels such as fiberboard and particleboard are predominantly manufactured using logging and processing residues, as well as low-quality wood like small stems (Sun et al., 2000). To meet the increasing demand for wood products and optimize the utilization of forest resources, it is common to use branches and barks in the production of wood-based panels. Branches possess physical and mechanical properties similar to those of the trunks and are often used as wood chips and fibers for particleboard and fiberboard production (Olarescu et al., 2022; Olarescu et al., 2023). Barks exhibit excellent acoustic and thermal insulation properties, making it suitable for manufacturing acoustic and insulation panels (Kain et al., 2020; Tudor et al., 2020).

**(b) In fact, table S2 "lumber" rate reflects such wood residues while making furniture/construction.**

The "lumber" rate ("wood output rate" in the revised manuscript) refers to the ratio of commercial wood output volume to wood logging volume. The commercial wood output volume refers to the volume of peeled wood that meets the national wood standards. In the wood processing (e.g., peeling and sawing) and transportation, there are several wood losses due to harvesting, processing, poor wood quality, and gathering transport (Local standard: DB35/T 1876-2019). Therefore, as we stated in section 2.3.2, what the wood output rate reflects is wood losses from wood logging volume to wood output volume, including poor quality wood, wood residues such as tree tops, barks, and sawdust, etc.

**3. (a) Wood-based panels, furniture, and construction typically are not mutually exclusive, which means some of the wood-based panels are used for furniture/construction. Can you check the definitions and make sure there is no double counting?**

In this study, the wood-based panels were not double counted with furniture and constructions. As we stated in section 2.2, the furniture pool only includes solid wooden furniture, and the constructions pool only refers to the structural components used to support buildings (e.g., beams and columns). However, the wood-based panels are usually used for building decoration (e.g., wall cladding and ceiling finishes) and panel furniture (non-solid wooden) manufacturing. The turnover time of wood-based panels is different from both solid wooden furniture and structural constructions (Table S3, Table S5). Therefore, there is no double counting. To clarify this point, we revised "furniture" as "solid wooded furniture" and "constructions" as "structural constructions" throughout the revised manuscript.

Due to the absence of statistical data on the quantity of roundwood utilized for plywood production post-2012, plywood was not individually quantified in our previous version. To clearly distinguish between wood-based panels and solid wooden furniture and structural constructions, we extracted the roundwood used for plywood production, which was allocated to the wood-based panels pool. From 2003 to 2012, roundwood used for plywood production comprised 6% to 12% of the total roundwood consumption. The proportion of roundwood used for plywood production post-2012 was replaced by the 2012 data. The results of carbon emissions and stocks of harvested carbon were revised accordingly.

**(b) I am also surprised that the wood-based panels are larger than the sum of furniture/construction.**

Wood-based panels holds a considerable share in the Chinese timber market. According to China Timber and Wood Products Distribution Industry Yearbook, fiberboard and particleboard comprised over 50% of domestically produced wood products in 2015 (China Timber and Wood Products Distribution Association, 2016). The production of solid wooden furniture relies on high-quality wood, which is primarily imported (China Timber and Wood Products Distribution Association, 2016). Consequently, the volume of solid wooden furniture produced from domestic wood is relatively low.

**4. What component in Fig.2/7 corresponds to the farmer's fuelwood?**

Farmer's fuelwood is included in the component of "fuelwood" in Fig.2/7. We revised Fig. 2 and clarified the fuelwood pool (refer to response #1).

**5. the data analysis and calculation steps are missing. The 2.2 data has fuelwood, pulpwood, wood-based panels, and sum of furniture/construction. The 2.3.2 data has roundwood, fuelwood, pulpwood, and noncommercial fuelwood.**

**(a) It is not clear whether roundwood in 2.3.2 includes fuelwood and pulpwood or not (as Fig.2).**

Sorry for the confusion. Roundwood includes pulpwood, not includes fuelwood (refer to revised Fig.2 in response #1).

**(b) Additionally, do fuelwood and pulpwood have the same values in these two data sources?**

Yes, the pulpwood has the same values in the section 2.2 and section 2.3.2, while the fuelwood in section 2.2 is the sum of commercial fuelwood and non-commercial fuelwood (i.e., farmers' fuelwood) (refer to response #1).

**(c) Based on "The carbon entering the residues pool can be calculated by subtracting the carbon in pools (1) to (5) from the total harvested carbon", there are several possibilities. For example: If roundwood includes fuelwood and pulpwood, then**

**residues pool =roundwood/R + noncommercial fuelwood – (fuelwood + pulpwood + wood-based panels + sum of furniture/construction).**

**If roundwood does not include fuelwood and pulpwood, and the two data sources have similar values for fuelwood/pulpwood, then**

**residues pool = (roundwood + fuelwood + pulpwood)/R + noncommercial fuelwood – (fuelwood + pulpwood + wood-based panels + sum of furniture/construction) = roundwood/R – wood-based panels – sum of furniture/construction + (fuelwood + pulpwood) * (1-R)/R + noncommercial fuelwood.**

**In the above formulas, R is a key parameter from a third source (NBS), and wood-based panels are the largest category (Fig.7). Please clarify the above calculation steps using explicit equations.**

Thank you for such thoughtful comment. We added an equation to clarify the allocation of harvested carbon to residues pool and revised the corresponding sentences in section 2.2.

"The wood in pools (1) to (5) was converted into carbon, with conversion factor of 0.229 for fuelwood and roundwood, and 0.269 for wood-based panels (IPCC, 2019b). Then, the carbon entering the residues ($Residue_C$) pool can be calculated by subtracting the carbon in pools (1) to (5) from the total harvested carbon (SHC in Sec. 2.3.2) as Eq. (7):"

$$Residue_C = SHC - \left( W_{output} + F_{log} \right) \times 0.229$$
$$-wood\_based\ panels \times 0.269 \qquad (7)$$

where *SHC (Eq. (21))* is the total harvested carbon for a given province, *Woutput* represents the volume of commercial wood output, and *Flog* represents the volume of non-commercial wood. Unlike commercial wood, the actual logging volume of non-commercial wood can be totally used effectively by farmers (Sec. 2.3.2). The sum of $W_{output}$ and $F_{log}$ is the total volume of pools (1) to (4), and wood-based panels here do not include plywood (Fig. 2)." (Lines 187-197 in the revised manuscript) (See more details in response #1)

**(d) discuss data uncertainties surrounding R.**

Thanks for your valuable comment. We have added a short discussion for the uncertainty of R.

"Wood output rate (R) is a key parameter for calculating the total wood harvest at provincial level. Its value varies depending on factors such as tree species, wood quality, and processing ways (Jiang et al., 2022). Even within the same province, R can vary significantly. However, due to the lack of R data at the sub-provincial level, this study utilized the provincial R values for 2009 obtained from the official website of National Bureau of Statistics (Table S2). Using provincial-level R for a single year overlooked the intra-provincial and inter-annual variations in R, potentially leading to bias in estimating the total wood harvest." Lines (528-534 in the revised manuscript)

**(e) I will be surprised if the residues pool will always be positive since R comes from an external independent source. In fact, the residues after 2014 are pretty small, which is quite suspicious, as we know fuelwood/residues are more than 60% globally (Fig.7).**

As we stated in sec. 2.2 and responded to comment #2(b), after harvesting, the killed understory vegetation and leaves are usually left on site to enhance soil fertility or treated as fuel, allocated to the residues pool. Meanwhile, not all branches and processing residues can be utilized, a portion of them also goes into the residue pool. Therefore, the residues pool must be positive. And the wood residues decreased from 2003 to 2018 demonstrates the increase in wood processing and utilization.

**Minor questions**

**6. The authors estimate that 80% of the harvested biomass is from selective logging. Assuming selective logging one pixel provides 50% of the biomass from clear-cutting a pixel, then the area**

**of selective logging will be 8x of the area of clear-cutting. Please provide the provincial level forest areas for clear-cutting and selective logging, respectively, in order to verify this result.**

Yes, it is a good idea to provide the provincial level forest areas for clear-cutting and selective logging. We added a figure that shows the number of pixels where clear-cutting and selective logging occurred and the proportion of harvested carbon from selective logging to the total harvested carbon at the provincial level averaged from 2003 to 2018. Besides, we added a short discussion in section 4.2.

"Nevertheless, selective logging has remained the principal way of forest harvesting in China. For entire China, the pixels occurred selective logging is about 50 folds of that occurred clear-cutting (Fig. S5), and the mean harvested biomass in a pixel from selective logging is 8% of that from clear-cutting (Fig. 6, Fig. S5). The occurrence of such a small percentage of biomass removal at pixel level suggests the ability of the LEAF dataset to capture minor disturbances." (Lines 469-473 in the revised manuscript)

[Figure]

**Figure S5:** The number of pixels where clear-cutting and selective logging occurred and the proportion of harvested carbon from selective logging to the total harvested carbon for each province averaged from 2003 to 2018.

**7. Figure 4: I still don't understand the response of comment #48 and I do not see any changes been made.**

**"48. Figure 4. It is hard to compare a and b as the scales in the legends are so different. It also**

**suggests that 30m resolution is less capable in capturing harvesting compared to the 0.1 degree."**

Sorry for the confusion. In several provinces (e.g., Xinjiang, Inner Mongolia, etc.), forest harvesting frequency is relatively low, resulting a poor visualization of forest harvesting in these regions at 30 m resolution (Fig. 4a). To enhance the visualization of spatial patterns of forest harvesting, we provided an additional map at 0.1° resolution (i.e., Fig. 4b). The unit of pixel value in Fig. 4a and 4b is 'g C $m^{-2}$ $yr^{-1}$', refers to the ratio of the annual total harvested carbon in a pixel to the corresponding pixel area. Not all of the 30 m pixels within the 0.1° × 0.1° range occurred harvesting, for those pixels without harvesting, the pixel value is 0. Therefore, the proportion of harvesting within the 0.1° pixel is much smaller than that within the 30 m pixel, despite the larger area of the 0.1° pixel compared to the 30 m pixel. As a result, the pixel value within 0.1° resolution is much smaller than that within 30 m resolution, resulting the scales in the legends of Fig. 4a and 4b are so different. As you stated, 0.1° resolution is more capable in capturing harvesting compared to the 30 m. The 0.1° graph we provided here is only to better demonstrate the spatial pattern of forest harvesting. Therefore, we did not make any change on Fig.4.

**References**

Sun, H., et al.: China Encyclopedia of Resources Science, Dongying: China University of Petroleum Press; Beijing: Encyclopedia of China Publishing House, 2020.

China Timber and Wood Products Distribution Association: China Timber and Wood Products Distribution Industry Yearbook, Beijing: China Building Materials Press, 2016.

Olarescu, A., Lunguleasa, A., and Radulescu, L.: Using deciduous branch wood and conifer spindle wood to manufacture panels with transverse structure, BioResources, 17(4), 6445-463, 2022.

Olarescu, A., Lunguleasa, A., Radulescu, L., and Spirchez, C.: Manufacturing and testing the panels with a transverse texture obtained from branches of Norway Spruce (*Picea abies L. Karst.*), Forests, 14, 665, 2023.

Kain, G., Tudor, E., and Barbu, M.: Bark thermal insulation panels: an explorative study on the effects of bark species, Polymers, 12, 2140, 2020.

Tudor, E., Dettendorfer, A., Kain, G., Barbu, M., Réh, R., and Krišt'ák, L'.: Sound-absorption coefficient of bark-based insulation panels, Polymers, 12, 1012, 2020.

DB35/T 1876-2019: A calculating method standard on output volume of main timber tree species. Fujian: Fujian Market Supervision Bureau.

IPCC: Refinement to the 2006 IPCC Guidelines for National Greenhouse Gas Inventories. Volume 4:

Agriculture, Forestry and Other Land Use. Chapter 12: Harvested Wood Products, Available at: https://www.ipcc-nggip.iges.or.jp/public/2019rf/pdf/4_Volume4/19R_V4_Ch12_HarvestedWoodProducts.pdf, 2019b, last access: 27 December 2023.

Jiang, X., Li, K., and Jiang, B.: Analysis on influencing factors of wood products blank outturn rate, Science Technology and Engineering, 22, 10930-10938, 2022.